



# Climatic signatures in early modern European grain harvest yields

Fredrik Charpentier Ljungqvist[1,2,3], Bo Christiansen[4], Jan Esper[5,6], Heli Huhtamaa[7,8],
Lotta Leijonhufvud[9,*], Christian Pfister[7,8], Andrea Seim[10,11], Martin Karl Skoglund[12], and Peter Thejll[4]

[1]Department of History, Stockholm University, 106 91 Stockholm, Sweden
[2]Bolin Centre for Climate Research, Stockholm University, 106 91 Stockholm, Sweden
[3]Swedish Collegium for Advanced Study, Linneanum, Thunbergsvägen 2, 752 38 Uppsala, Sweden
[4]Danish Meteorological Institute, Lyngbyvej 100, 2100 Copenhagen Ø, Denmark
[5]Department of Geography, Johannes Gutenberg University, 551 28 Mainz, Germany
[6]Global Change Research Institute (CzechGlobe), Czech Academy of Sciences, 603 00, Brno, Czech Republic
[7]Institute of History, University of Bern, 3012 Bern, Switzerland
[8]Oeschger Centre for Climate Change Research, University of Bern, 3012 Bern, Switzerland
[9]Department of Historical Studies, University of Gothenburg, 405 30 Gothenburg, Sweden
[10]Chair of Forest Growth and Dendroecology, Institute of Forest Sciences, Albert Ludwig University of Freiburg, 791 06 Freiburg, Germany
[11]Department of Botany, University of Innsbruck, 6020 Innsbruck, Austria
[12]Division of Agrarian History, Department of Urban and Rural Development, Swedish University of Agricultural Sciences, 750 07 Uppsala, Sweden
[*]Current affiliation: Upplands-Bro Municipality, 196 81 Kungsängen, Sweden

**Correspondence:** Fredrik Charpentier Ljungqvist (fredrik.c.l@historia.su.se)

**Abstract.** The association between climate variability and grain harvest yields has been an important component of food security and economy in European history. Yet, inter-regional comparisons of climate–yield relationships have been hampered by locally varying data types and use of different statistical methods. Using a coherent statistical framework, considering the effects of diverse serial correlations on significance, we assess the temperature and hydroclimate (precipitation and drought) signatures in grain harvest yields across varying environmental settings of early modern (*c.* 1500–1800) Europe. An unprecedentedly large network of yield records from northern (Sweden), central (Switzerland) and southern (Spain) Europe is compared with a diverse set of seasonally and annually resolved palaeoclimate reconstructions. Considering the effects of different crop types and time-series frequencies, we find within regions consistent climate–harvest yield patterns characterised by a significant summer soil moisture signal in Sweden, a winter temperature and precipitation signal in Switzerland, and spring and annual mean temperature signals in Spain. The regional scale climate–harvest associations are weaker than the recently revealed signals in early modern grain prices, albeit similar to those observed in modern climate–harvest relationships on comparable spatial scales.

## 1 Introduction

Agriculture has been, and still is, highly weather and climate dependent (Hoogenboom, 2000; Cantelaube and Terres, 2005; White et al., 2018; Vogel et al., 2019; Heino et al., 2020). In pre-industrial Europe, when human livelihood mainly relied on grain production for calorie intake (Allen, 2000), climate variations – and associated changes in weather patterns – were of



immense importance for society (Degroot et al., 2021; Pfister and Wanner, 2021; Ljungqvist et al., 2022). Stimulated by the want to place contemporary global warming impacts on agriculture into a long-term perspective, investigations of past climate–harvest yield relationships have attained an increasing interest during the past two decades (for a review, see Ljungqvist et al.,

2021). However, much of this scholarship has focused on the consequences for food security, and related societal impacts, in case of consecutive years of adverse climatic conditions for grain production (Pfister and Brázdil, 1999; Pfister, 2005; D'Arrigo et al., 2020; Huhtamaa et al., 2022; Stoffel et al., 2022). Years and decades with cold springs, and excessively wet summers, were associated with unfavourable conditions for crop growth north of the Mediterranean region (Camenisch, 2015; White et al., 2018; Pfister and Wanner, 2021). Droughts could have severe negative impacts on grain harvest yields too (Esper et al.,

2017; Brázdil et al., 2019, 2020; Skoglund, 2022; Związek et al., 2022), although their agricultural impacts were geographically heterogeneous (Wetter et al., 2014; Brázdil et al., 2019) as is still the case in Europe today (Beillouin et al., 2020). In general, droughts are more spatially restricted than temperature anomalies in Europe (Büntgen et al., 2010; Cook et al., 2015; Ljungqvist et al., 2019), and they are less likely than temperature anomalies to occur for consecutive years (Bunde et al., 2013; Franke et al., 2013; Esper et al., 2017).

Quantitative and semi-quantitative analyses to estimate the impact of climate variability on past grain harvests extend back to the beginning of the nineteenth century (Herschel, 1801), followed by studies by Brückner (1895), Beveridge (1921, 1922), Le Roy Ladurie (1967), Parry (1975, 1976, 1978), and Pfister (1979, 1984, 1988). Despite a comprehensive body of scholarship, conducted across different disciplines, only a handful of studies has attempted to statistically quantify the effect of temperature and hydroclimate (i.e., wetness or dryness) on harvest yields of different grain types in early modern Europe. Recent advances

in palaeoclimatology have facilitated this research (Ljungqvist et al., 2021), although large-scale studies or inter-regional comparisons remain rare. An exception is the study by Pei et al. (2016) who, using yield ratio data from Slicher van Bath (1963), demonstrated a regionally different recovery after the 'climax' of the Little Ice Age *c.* 1570–1710 (Wanner et al., 2022): yields increased well above their pre-1570 level in western Europe as opposed to eastern Europe. Those differences were due to socio-political and technological factors rather than climate (Ljungqvist et al., 2021). Most longitudinal studies

have detected relatively weak, and over time unstable, climate–grain harvest relationships.

The difficulty in obtaining time-series of harvest yields from early modern Europe (Section 2.1) has resulted in the frequent use of grain prices as a substitute for actual harvest data. Significant effects of climate variability on grain prices at inter-annual time-scales has long been established (see, e.g., Mauelshagen, 2010). However, the influence of climate variability on decadal and longer time-scales still remains contested. This is, to a large part, a result of that price inflation (and deflation) needs to be

removed from the price series which consequently removes long-term price changes too. By applying spline filter techniques to emphasise common mid-frequency variability, Esper et al. (2017) and Ljungqvist et al. (2022) found strong negative grain price–temperature relationships (i.e., colder = high prices and *vice versa*) across Europe, which are of episodic rather than periodic (cyclic) nature (Ljungqvist et al., 2022). These studies showed that temperature strongly influenced early modern European grain production, and thus food security, to a larger extent than previously established.

Existing studies of climate effects on historical grain yields have mainly focused on very cold periods, in particular those containing known episodes of famines (e.g., Camenisch et al., 2016; D'Arrigo et al., 2020; Huhtamaa et al., 2022), or on



extreme droughts (e.g., Brázdil et al., 2019) or floods (e.g., Kiss, 2019). However, a number of studies have also investigated the relationship between climate variability and grain yields over longer time periods (e.g., Landsteiner, 2005; Pfister, 2005; Edvinsson et al., 2009; Campbell, 2010, 2016; Huhtamaa et al., 2015; Huhtamaa and Helama, 2017; Skoglund, 2022). These

studies have broadly shown that in northern Europe, at locations close to the thermal limits for grain agriculture, growing season length and temperature determine grain yields (for a review, see Ljungqvist et al., 2021). However, in the main agricultural areas of south-eastern and southern Sweden spring and summer hydroclimate conditions were the most important climate drivers of grain yields. Particularly, maximum harvest yields were obtained after warm winters and springs followed by cool and relatively moist summers (Edvinsson et al., 2009; Holopainen et al., 2012; Ljungqvist and Huhtamaa, 2021; Skoglund,

60  2022).

In Central Europe, including Switzerland, cold spring periods during March–April and excessive summer precipitation lowered the grain harvest (Pfister, 2005), while moist conditions during September reduced the sown acreage and resulted in washed-out soil nitrogen (Pfister, 2007). In particular for autumn-sown rye, wet autumns and winters had adverse effects on next season's yields (Pfister, 1988) and on seed quality (Landsteiner, 2005). Drought had locally varying impacts on grain

yields (Wetter et al., 2014; Brázdil et al., 2019). Most importantly, clusters of rainy autumns, cold springs and wet mid-summers could led to multiple consecutive harvest failures and threatened food security across large regions of Central Europe (White et al., 2018; Pfister and Wanner, 2021). Similar climatic signatures in the grain harvests, but with an even larger sensitivity to wet summer conditions, have been identified in maritime regions of western Europe, including England, where high July–August temperatures also tended to increase yields (Scott et al., 1998; Michaelowa, 2001; Brunt, 2015; Campbell, 2016;

Pribyl, 2017; Soens, 2022). In Switzerland, and Central Europe in general, spelt, wheat, rye and some oats were autumn-sown (in September and October) while all barley and some oats were spring-sown (in March and April depending on the altitude). The spring-sown crops were harvested at the end of May or during June and autumn-sown crops were harvested from mid-July onwards with exact timing dependent on both altitude and year (Pfister, 1984).

Due to the complex topography and diverse local-scale climatic conditions, the Mediterranean region, including Spain,

exhibits a highly variable climate between regions (Rodrigo and Barriendos, 2008; Santiago-Caballero, 2013b; Llopis et al., 2018, 2020). Many drought-prone regions were historically characterised by low average yields (Santiago-Caballero, 2013a). While spring droughts were generally the dominant climatic threat (Barriendos, 2005; Llopis et al., 2020) adverse effects might also arise from cold winter and/or springs (Moreda, 2017; Izdebski et al., 2018; Moreno et al., 2020). Autumn precipitation could have an effect on harvest yields as it provided soil moisture for the subsequent growing season. Wheat was the main

crop in early modern Spain, sown in October to November and harvested in June in the south and July in the north, and mainly cultivated in the basins of major rivers (Moreno et al., 2020). Because of the warm climate and regular summer droughts, the main grain growing season in Spain extended from November to May, rendering summer climate conditions largely irrelevant for grain crops (Simpson, 1996).

Similar to what has been found for historical times, only relatively weak climate–yield relationships are reported for twen-

tieth and twenty-first century European data. For example, Trnka et al. (2016) demonstrated rather weak climate–yield relationships for barley and wheat across most of Europe during the 1901–2012 period. While barley often showed insignificant



relationships to monthly or seasonal temperature and precipitation variability, wheat showed stronger relationships. The explained variance from a *combination* of climate variables over the April–June season exceeded 25% for wheat for Belgium, Bulgaria, Lower Saxony (Germany), Sweden, and Tuscany (Italy), although it did not exceed 13% on average for Europe (Trnka

et al., 2016). Climatic impacts explain also only a limited proportion of the modern German year-to-year grain yield variability (Albers et al., 2017). For much of Europe, drought conditions during April–June have the greatest impact on crop yields as demonstrated for, e.g., the Czech Republic (Hlavinka et al., 2009) and Belgium (Gobin, 2012). Beillouin et al. (2020) found stronger relationships, explaining on average 34% of the harvest variations for a variety of crops across Europe, with 46% and 15%, respectively, of the variance in yields explained by climate in northern/western and southern Europe since the early

twentieth century.

In this article, we evaluate the association between grain harvest yields and seasonal temperature and hydroclimate variations in northern, central and southern Europe during the 1500–1800 period. We compare data from Sweden, Switzerland and Spain – as well as from some other regions such as England and southern Germany – and analyse how the yields of different grain types were impacted by varying climatic conditions. The assessment of sites across a north–south transect including diverse

environmental settings supports comparisons of regional temperature and hydroclimatic impacts on different grain species. This study is the first to systematically assess the strength of the climate signal, and its significance, in early modern Swedish, Swiss and Spanish harvest data using a range of climate reconstructions.

Considering previous work on modern and historical climate–harvest relationships, we hypothesise for Sweden that warm winters and springs, relatively cool summers, and moderately high precipitation in spring and early summer benefited grain

yield, whereas cold and/or dry springs and summers limited yields. Grain yields in Switzerland are expected to benefit from dry autumns and winters, warm springs and dry mid-summers. For Spain, wet and warm winters and springs are expected to increase grain yields, whereas summer conditions are irrelevant to the grain harvest because of the early harvest date. Following Ljungqvist et al. (2022), we emphasise conservatively calculated significances to establish the climate–harvest relationships. Additionally, we test for causality, i.e., the directions between the influence of one variable on another (e.g., climate

variability causes changes in harvest yields).

## 2 Materials and methods

### 2.1 Grain harvest data

Three main types of harvest data exist for the early modern period in Europe: (1) tithe records from areas where the tithe followed actual harvest size variations, (2) harvest yield ratio (i.e., sown-to-harvested grain) records from individual estates

and farms, and (3) total yield quantity records for individual estates and farms or larger districts. We have collected and digitised an unprecedentedly large dataset from published literature of both tithe and yield series of which only the longest (near) continuous ones have been included in this study (Tables 1 and 2). Whereas tithe data represent (sub)regional harvest variations, as the tithe data are recorded on the parish level, the yield data typically reflect local (estate) scale yield variations. Tithe records, as they represent the output from numerous farms and usually cover a larger geographical area than yield series,

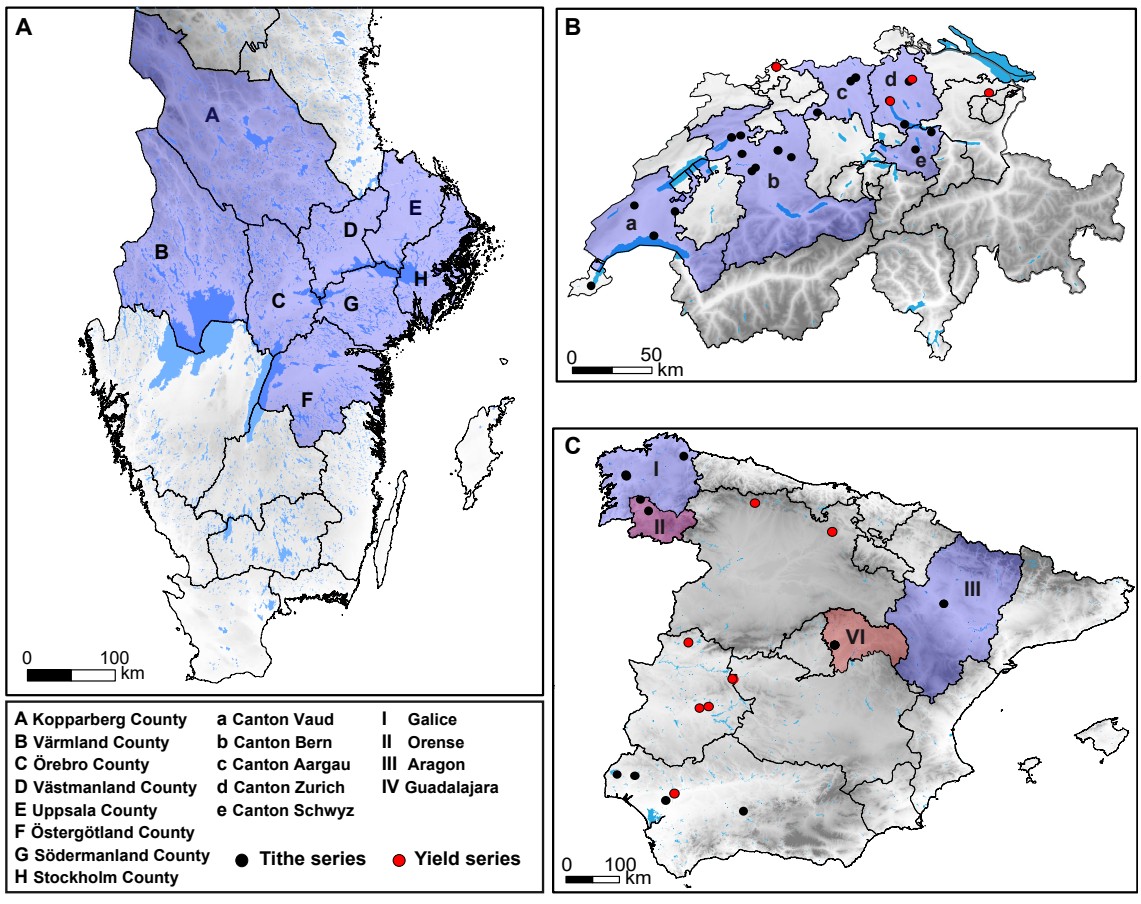

**Figure 1.** Origin of grain tithe and yield data from **A** Sweden, **B** Switzerland, and **C** Spain. Note that for Sweden we use county-level average tithe data for barley/rye, oats, and wheat, respectively, and for Switzerland canton-level averages of all grain types, whereas the tithe data for Spain derives from differently-sized administrative districts. Due to their short length, and many data gaps, no yield series are used for Sweden.

can be expected to be less biased by individual outliers of single farms. The potential to study climate signals in early modern harvests using tithe data was already noted by Stauffer and Lüthi (1975). Continuous grain tithe data exist from parts of early modern Sweden, Switzerland and Spain as well as for some individual locations elsewhere across Europe (Table 1). However, yield series (for example from one farm) may not even be locally representative. The effect of local soil conditions will be considerable, and microclimate and local weather events (e.g., one severe thunderstorm or local radiation night frost) will

affect the yield ratio as well as the total yield quantity.

The lag-1 auto-correlation is much stronger in the tithe data compared to the yield data (mean AR1 = 0.59 *vs*. 0.27) (Tables 1 and 2), indicating that the tithe time-series contain a much larger 'memory' from one year to the next. This pattern holds true for all study regions albeit with large variations between individual tithe and yield series within each region. As a comparison,





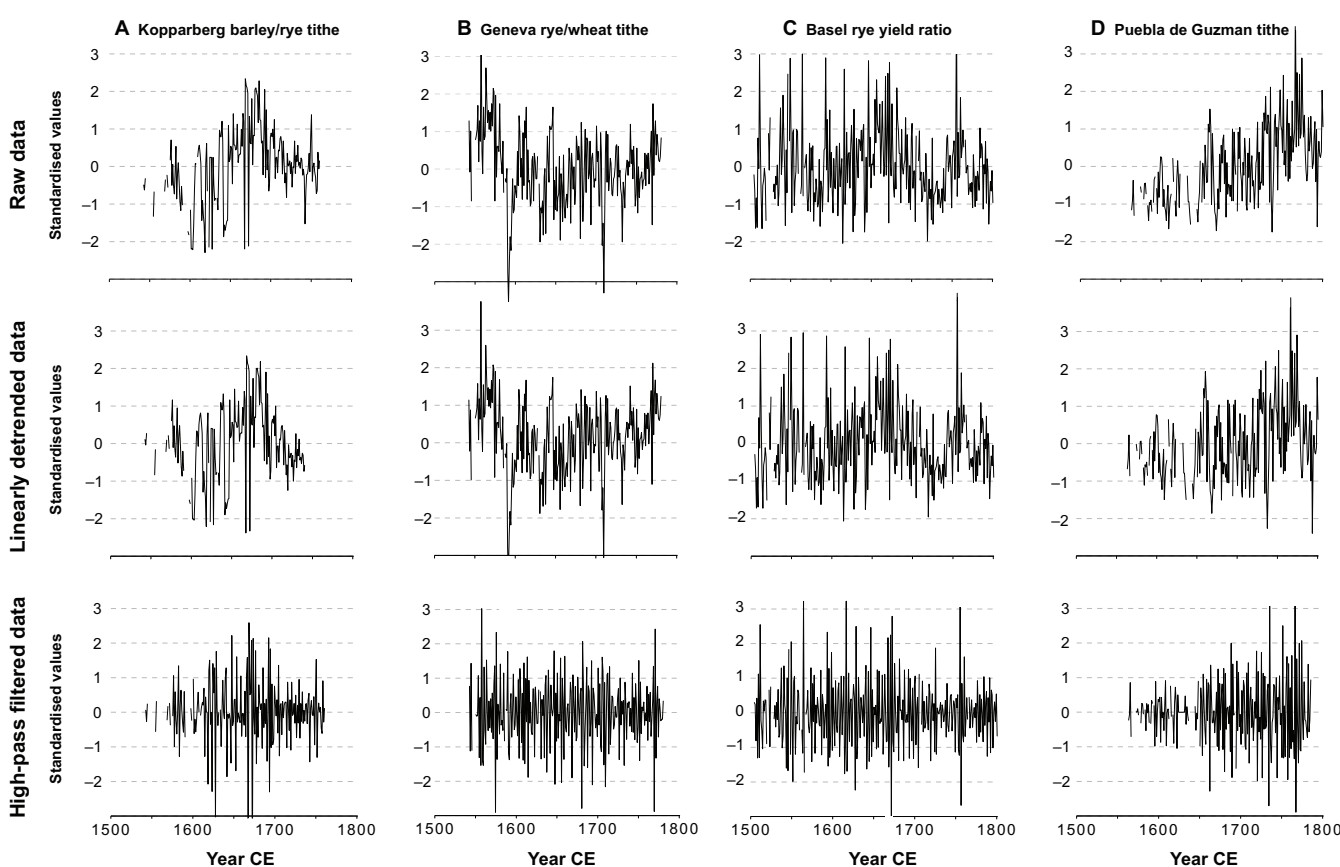

**Figure 2.** Examples of harvest yield data from **A** Sweden (Kopparberg, barley/rye tithe), **B** Switzerland (Geneva, rye/wheat tithe and Basel rye yield ratio), and **C** Spain (Puebla de Guzman tithe) for non-detrended ('raw'), linearly detrended, and 10-year Gaussian high-pass filtered data. All the series shown have been standardised to zero mean and standard deviation of one. Note that the difference between non-detrended and linearly detrended is small in some series (e.g., Geneva) but relatively large in others (e.g., Puebla de Guzman).

grain price data contains an even stronger auto-correlation (mean AR1 = 0.78) (Table A1), presumably due to the effects of

storage, trade and price regulations (Rahlf, 1996; Persson, 1999; Alfani and Ó Gráda, 2017; Ljungqvist and Seim, 2023). Temperature and hydroclimate reconstructions from documentary data contain auto-correlations similar to yield data, whereas tree-ring based climate reconstructions instead contain auto-correlations similar to tithe data (Table 3; see further Section 4.1).

     For Sweden, the tithe data is only used from 1572 as the data from that year onwards were obtained through modern archival research (Myrdal and Söderberg, 1991; Leijonhufvud, 2001; Hallberg et al., 2016), whereas data prior to 1572 were

compiled by Leijonhufvud (2001) from the publication by Forssell (1872), who's criteria and methods for collection remain uncertain. We merged the Swedish tithe data for barley and rye into a single 'barley/rye' category as the two grain types are frequently reported as just 'grain' in the tithe accounts (Ågren, 1964). In principle, 'grain' in accountings in general





implied fifty-fifty between rye and barley (Jansson et al., 1993), though this proportion shifted in reality (Leijonhufvud, 2001).
Owing to insufficient tithe data coverage in some counties, we truncated the end date of individual county-level averaged tithe
series (i.e., Östergötland and Södermanland at 1680, Stockholm and Värmland 1730, Uppsala and Västmanland 1740, and
Kopparberg 1760). Reliable tithe data for Örebro County is restricted to 1665–1720. We have *not* for this study included any
of the tithe data series from northern Sweden, present-day Finland or from the former Danish and Norwegian provinces that
became Swedish in the mid-seventeenth century. Tithe data from Skaraborg County was also excluded as it contains numerous
missing values (Leijonhufvud, 2001) and has a more maritime climate than the other counties (Wastenson et al., 1995). For
Switzerland, rye and wheat production for the Republic of Bern and four districts of the Republic of Zurich have been estimated
back to *c.* 1530 based on tithe receipts (Pfister, 1984). Between 1755 and 1797, aggregate tithe receipts represented 57% of
the estimated total grain harvest of which spelt accounting for the majority (see Tables 26 and 27 in Pfister, 1984). The tithe
series for Spain are more heterogeneous with respect to quality and spatial domain (Ponsot, 1969; Eiras Roel, 1982; Ciria,
2007). Much of the tithe data are not separated by grain type, but reflect the total agricultural productivity of grain, wine, and
often also vegetables (Le Roy Ladurie and Goy, 1982). Only longer and more continuous tithe series for mainland Spain have
therefore been included (Table 1).

     The majority of the published yield ratio series or area yield data series are composed of short data series, frequently with
numerous gaps (e.g., Slicher van Bath, 1963; Tornberg, 1989; Leijonhufvud, 2001; Young, 2007). Such yield ratio series are
not included in this study as they complicate assessments of statistically significance in climate–harvest yield relationships.
Yield data, sufficient in length for performing correlation analyses against (palaeo)climate data, exist for Switzerland and
Spain (although only for the eighteenth century in the latter case) as well as for some individual locations elsewhere in Europe
(Table 2). Both tithe data and yield data are henceforth referred to as 'grain harvest data' or just 'harvest data'.

## 2.2   Temperature and hydroclimate data

Instrumental measurements of temperature are available for portions of Europe since the early eighteenth century, albeit the
longest extend back to 1659, and increasingly from the mid-eighteenth century onwards for precipitation (Jones, 2001; Briffa
et al., 2009; Brönnimann et al., 2019). Prior to this instrumental period indirect, 'proxy', climate information has to be de-
rived from either historical documentary sources (e.g. Pfister and Wanner, 2021) or natural archives such as tree rings (e.g.
Cook et al., 2015). Seasonal gridded reconstructed temperature data – for winter, spring, summer and autumn – covering
Europe is available since 1500 (Luterbacher et al., 2004; Xoplaki et al., 2005) – henceforth referred to as Luterbacher et al.
(2004) (Table 3). These gridded products combine instrumental measurements from the eighteenth century onwards with doc-
umentary and natural proxy data. We also use the summer (June–August) temperature data from Luterbacher et al. (2016), as
updated by Ljungqvist et al. (2019), derived from quality-screened temperature-sensitive tree-ring data and historical docu-
mentary evidence for June–August from Dobrovolný et al. (2010) resolved at a $5° \times 5°$ grid. For the Luterbacher et al. (2004)
and Ljungqvist et al. (2019) reconstructions, the local grid cells for each study region has been extracted, averaged and utilised
for the correlation analyses. In addition, we use for Switzerland the seasonal documentary-based seasonal resolved temperature



**Table 1.** Information of the grain tithe data including the covered period and missing values ('gaps' in percentage), the grain type(s) (B = barley; R = rye; O = oats; W = wheat; B/R = barley/rye mixture; R/W = rye/wheat mixture), the auto-correlation coefficient for lag 1 year, AR1 (numbers in the order of the listed grain types) and the data source(s).

| Location | Period | Grain type(s) | Gaps | AR1 | Source(s) |
|---|---|---|---|---|---|
| *Sweden* (8 counties) | | | | | |
| Kopparberg County | 1572–1740 | B/R,O,W | 0.59, 20.71, 50.90 | 0.62, 0.93, 0.76 | Leijonhufvud (2001); Hallberg et al. (2016) |
| Örebro County | 1665–1720 | B/R,O,W | 0, 7.14, 7.14, | 0.28, 0.37, 0.50 | Leijonhufvud (2001); Hallberg et al. (2016) |
| Östergötland County | 1573–1680 | B/R,O,W | 12.04, 13.46, 11.54 | 0.58, 0.94, 0.73 | Leijonhufvud (2001); Hallberg et al. (2016) |
| Södermanland County | 1572–1681 | B/R,O,W | 17.27, 26.61, 26.61 | 0.23, 0.45, 0.49 | Leijonhufvud (2001); Hallberg et al. (2016) |
| Stockholm County | 1572–1730 | B/R,O,W | 27.67, 30.82, 32.06 | 0.54, 0.70, 0.89 | Leijonhufvud (2001); Hallberg et al. (2016) |
| Uppsala County | 1572–1740 | B/R,O,W | 28.99, 39.34, 36.09 | 0.55, 0.58, 0.91 | Leijonhufvud (2001); Hallberg et al. (2016) |
| Värmland County | 1575–1730 | B/R,O,W | 13.13, 15.48, 46.85 | 0.36, 0.83, 0.75 | Leijonhufvud (2001); Hallberg et al. (2016) |
| Västmanland County | 1572–1740 | B/R,O,W | 36.69, 7.69, 7.69 | 0.67, 0.61, 0.60 | Leijonhufvud (2001); Hallberg et al. (2016) |
| | | | | | |
| *Switzerland* (19 series) | | | | | |
| Burgdorf | 1550–1825 | R/W | — | 0.77 | Pfister (1984) |
| Cappelerhof | 1531–1797 | R/W | 0.37 | 0.63 | Pfister (1984) |
| Fraumünsteramt | 1531–1797 | R/W | — | 0.48 | Pfister (1984) |
| Frienisberg | 1529–1797 | R/W | 0.74 | 0.63 | Pfister (1984) |
| Geneva | 1542–1780 | R/W | 2.51 | 0.38 | Head-Köenig and Veyrassat-Herren (1972) |
| Gottstatt | 1558–1825 | R/W | — | 0.32 | Pfister (1984) |
| Königsfelden | 1556–1797 | R/W | — | 0.35 | Pfister (1984) |
| Köniz | 1732–1825 | R/W | — | 0.29 | Pfister (1984) |
| Lausanne | 1538–1796 | R/W | 1.54 | 0.88 | Pfister (1984) |
| Moudon | 1564–1796 | R/W | 2.15 | 0.88 | Pfister (1984) |
| Nidau | 1535–1825 | R/W | — | 0.67 | Pfister (1984) |
| Romainmôtier | 1538–1796 | R/W | 1.54 | 0.67 | Pfister (1984) |
| Spital | 1700–1797 | R/W | 1.02 | 0.67 | Pfister (1984) |
| Stift Bern | 1529–1796 | R/W | 0.74 | 0.88 | Pfister (1984) |
| Töss | 1529–1797 | R/W | 0.37 | 0.37 | Pfister (1984) |
| Trachselwald | 1534–1797 | R/W | 4.91 | 0.84 | Pfister (1984) |
| Wädenswil | 1533–1797 | R/W | 0.75 | 0.72 | Pfister (1984) |
| Wangen | 1686–1825 | R/W | — | 0.54 | Pfister (1984) |
| Zofingen | 1542–1797 | R/W | — | 0.38 | Pfister (1984) |
| | | | | | |
| *Spain* (10 series) | | | | | |
| Mondoñedo | 1595–1800 | Mixed | 0.10 | 0.96 | Eiras Roel (1982) |
| Santiago | 1606–1799 | Mixed | 4.12 | 0.76 | Eiras Roel (1982) |
| Galice | 1595–1800 | Mixed | — | 0.82 | Eiras Roel (1982) |
| Orense | 1598–1800 | Mixed | — | 0.84 | Eiras Roel (1982) |
| Aragon | 1660–1827 | W | 29.84 | 0.34 | Ciria (2007) |
| Guadalajara | 1700–1800 | B,R,O,W | — | 0.36, 0.71, 0.62, 0.17 | Santiago-Caballero (2014) |
| Calañas | 1563–1835 | W | 11.35 | 0.36 | Ponsot (1969) |
| Puebla de Guzman | 1563–1835 | W | 11.35 | 0.43 | Ponsot (1969) |
| Lucena | 1563–1835 | W | 11.35 | –0.08 | Ponsot (1969) |
| Bollullos | 1563–1835 | W | 10.92 | 0.34 | Ponsot (1969) |
| | | | | | |
| *Other regions* (4 series) | | | | | |
| Beaume, France | 1597–1788 | O/W | — | 0.42 | Goy (1972) |
| Nurenberg, Germany | 1500–1670 | R | — | 0.75 | Bauernfeind (1993) |
| Alsace, France | 1501–1790 | Mixed | — | 0.88 | Veyrassat-Herren (1972) |
| French Mediterranean | 1532–1788 | Mixed | 2.33 | 0.43 | Goy and Head-Köenig (1969) |





**Table 2.** Information of the grain yield series including location/site, the covered period and missing values ('gaps' in percentage), the grain type(s) (B = barley; R = rye; O = oats; S = spelt; W = wheat), the auto-correlation coefficient for lag 1 year, AR1 (numbers in the order of the listed grain types) and the data source.

| Location | Period | Type(s) | Gaps | AR1 | Source(s) |
|---|---|---|---|---|---|
| *Switzerland* (7 series) | | | | | |
| Basel | 1500–1800 | R,S | 4.04, 3 | 0.08, 0.21 | Head-Köenig (1979) |
| St. Gallen | 1549–1799 | S | 5.58 | 0.19 | Head-Köenig (1979) |
| Zurich | 1501–1800 | R,S | 10.33, 3.69 | 0.28, 0.63 | Head-Köenig (1979) |
| Winterthur | 1536–1785 | R,S | 16, 15.26 | 0.15, 0.24 | Head-Köenig (1979) |
| | | | | | |
| *Spain* (8 series) | | | | | |
| El Rincón farm | 1699–1781 | W | — | 0.03 | Llopis et al. (2020) |
| La Burguilla farm | 1699–1781 | W | — | –0.06 | Llopis et al. (2020) |
| Madrigalejo farm | 1699–1781 | W | 12.88 | 0.16 | Llopis et al. (2020) |
| La Vega farm | 1699–1781 | W | — | 0.16 | Llopis et al. (2020) |
| Quintanajuar farm | 1699–1781 | W | 5.88 | 0.48 | Llopis et al. (2020) |
| Panguia farm[a] | 1695–1799 | W | — | –0.07 | Llopis et al. (2020) |
| Farms of Matallana | 1699–1781 | W | — | 0.10 | Llopis et al. (2020) |
| Rinconada Alta farm[a] | 1695–1799 | W | — | –0.05 | Llopis et al. (2020) |
| | | | | | |
| *Other regions* (5 series) | | | | | |
| Arles, France | 1621–1787 | W | 22.75 | 0.04 | Goy (1972) |
| England (average)[a] | 1551–1800 | B,O,W | — | 0.97, 0.98, 0.55 | Broadberry et al. (2015) |
| Siena, Italy[a] | 1546–1667 | W | 12.30 | 0.23 | Parenti (1942) |

[a] Yield per unit area and not yield ratio.

reconstruction of Dobrovolný et al. (2010) and for Sweden the documentary-based January–April temperature reconstruction of Leijonhufvud et al. (2010).

A reconstruction of gridded seasonal precipitation estimates are available since 1500 for Europe (Pauling et al., 2006), although it has limited skill prior to the early eighteenth century (Section 4.5). As an approximation of growing season soil
moisture (drought) conditions we use the Old World Drought Atlas (Cook et al., 2015). This benchmark tree-ring based reconstruction provides June–August soil moisture conditions as self-calibrated Palmer Drought Severity Index (scPDSI) values on a $0.5° \times 0.5°$ grid across Europe. The atlas has limited skill in northern (Ljungqvist et al., 2019) and eastern (Cook et al., 2020) Europe; furthermore, it appears to somewhat underestimate the amplitude of low-frequency (multi-decadal to centennial scale) hydroclimatic variations at regional scales (compared with, e.g., Scharnweber et al., 2019; Büntgen et al., 2021). The local

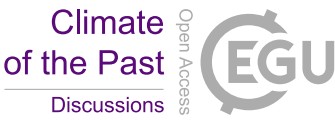

**Table 3.** Information of the palaeoclimate series including the location, climate parameter, season, data type, the auto-correlation coefficient for lag 1 year, AR1, and data source.

| Dataset | Season | Type | AR1 | Source |
|---|---|---|---|---|
| *Temperature* | | | | |
| Entire Europe (gridded) | June–August | Tree-ring (MXD/TRW) and doc. | 0.59[a] | Ljungqvist et al. (2019) |
| Lötschental | June–August | Tree-ring (MXD) | 0.59 | Büntgen et al. (2006) |
| Central Europe | Seasonal | Doc. | 0.16 | Dobrovolný et al. (2010) |
| Temperature (gridded) | Seasonal | Multi-proxy/instr. | 0.06[a] | Luterbacher et al. (2004) |
| Stockholm[c] | JFMA | Doc. | 0.17 | Leijonhufvud et al. (2010) |
| Pfister Switzerland indices | Seasonal | Doc. | 0.21[b] | Pfister (1992) |
| | | | | |
| *Hydroclimate data* | | | | |
| Old World Drought Atlas (scPDSI, gridded) | JJA | Tree-ring (MXD/TRW)[d] | 0.36[a] | Cook et al. (2015) |
| Scandinavia SPEI | May–June[e] | Tree-ring (TRW) | 0.57 | Seftigen et al. (2017) |
| Upper Rhine groundwater level | Annual | Tree-ring (TRW) | 0.80 | Tegel et al. (2020) |
| S. Germany SPEI | AMJJA | Tree-ring (TRW) | 0.69 | Muigg et al. (2020) |
| Precipitation (gridded) | Seasonal | Multi-proxy/instr. | 0.19[b] | Pauling et al. (2006) |
| Pfister Switzerland indices | Seasonal | Doc. | 0.11[b] | Pfister (1992) |

[a] Mean AR1 values of the area means of the different study regions.

[b] Mean AR1 values of the different seasonal means.

[c] Starts 1502 and with some gaps in the early sixteenth century.

[d] Only one out of 106 predictors are MXD. The rest are TRW.

[e] June SPEI aggregated over two months' window, which corresponds to cumulative precipitation and evapotranspiration over May and June.

grid cells from the Pauling et al. (2006) and Cook et al. (2015) reconstructions for each study region has been extracted and utilised for the correlation analyses. For particular regions, additional non-gridded tree-ring based hydroclimate reconstructions are used. This includes an annual groundwater level reconstruction from the Upper Rhine River Valley (Tegel et al., 2020), an April–August Standardised Precipitation-Evapotranspiration Index (SPEI) reconstruction from Bavaria, Germany (Muigg et al., 2020), and a June SPEI reconstruction from Scandinavia (Seftigen et al., 2017). For Switzerland, we used the Pfister

(1992) indices, ranging from $+3$ (extremely warm, respectively wet) to $-3$ (extremely cold, respectively dry), derived from historical weather descriptions (Adamson et al., 2022). Index values of $-1$, 0, and 1 characterise rather average seasons derived from only narrative evidence, whereas values $> 1$ and $< -1$ integrate both narrative and proxy evidence (for a further description of the Pfister indices, see Pfister and Wanner, 2021).





### 2.3   Detrending, correlation analysis and significance

We detrended the harvest data because long-term yield trends are heavily influenced by factors related to human agency as opposed to climatic drivers (see Section 4.7). Two different detrending methods are used: (1) linear detrending preserving variability on up to multi-decadal time-scales, and (2) 10-year high-pass Gaussian filtering preserving variability on only sub-decadal time-scales (Fig. 2). All climate series have been detrended the same way as the harvest series. Linear detrending was performed by removing from the series an ordinary least squares fit between the series and time. The linear fit is found

by minimising the mean square error. A possible issue with using linear detrending of early modern grain harvest data is the absence of a non-ambiguous long-term linear trend in some datasets. For example in Sweden (Leijonhufvud, 2001; Hallberg et al., 2016) and in several regions of Switzerland (Pfister, 1984), it is difficult to discern any clear long-term productivity increases prior to the late eighteenth century. The trends in these series are often difference-stationary, since there are stochastic shifts in the mean (Bauernfeind and Woitek, 1996). Tithe and yield series from some parts of Spain do appear to show long-term

productivity gains earlier (Santiago-Caballero, 2013b), resulting in a larger effect on linear detrending (Fig. 2). While missing data are unproblematic for the linear detrending they might affect the 10-year high-pass Gaussian filter. Before applying the filter, we therefore impute the missing values with a kernel smoothing method (Hastie et al., 2001). After applying the Gaussian filter, we restored the status of the missing values. We have also tested other detrending procedures as alternative to 10-year high-pass Gaussian filtering – e.g., first differencing – and find that the result is relatively robust to this choice. After detrending,

we standardised the data to produce series with a mean of zero and standard deviation of one. For calculating regional averages (e.g., all Swiss yield ratio series for rye), we lined the segments up according to time and then produced an aggregate series by averaging the values. We calculated no average for Spanish tithe or yield data given the complex topography and heterogeneous climatic conditions in Spain (Section 1).

   In this study we quantify the association between the time-series by the Pearson correlation coefficient. The finite length

of the time-series means that spurious correlations can appear by chance. This problem is aggravated by the presence of considerable serial correlations in time-series as evidenced by their rather large AR1 coefficients (see Tables 1– 3). It is therefore important to report the statistical significance of correlations. The significance is calculated, as in Ljungqvist et al. (2022), both by the traditional method based on the Student's $t$-statistics and by a phase-scrambling based method. The latter method takes the actual number of degrees of freedom into account (Schreiber and Schmitz, 2000) and gives more accurate results

when the time-series contain serial correlations. This method is therefore more strict and report fewer spurious significant correlations than the $t$-test (Bartlett, 1935; Ljungqvist et al., 2022). Thus, we focus on results obtained with the conservative phase-scrambling-based significance test although, for comparison, we also report significance obtained from the traditional $t$-test.

   It should be noted that in some cases we find weak and marginally significant correlations. However, if such correlations

are found in several related pairs of time-series – such as correlations between yields in neighbouring regions and different but related climate indices – the belief in a physical (non-spurious) connection is increased. Obtaining a formal significance level in this situation is difficult, instead we here adopt an informal manner, choosing to emphasise those results that individually



are somewhat marginal, but which taken together paint a consistent picture. Time-dependent variance in one or both of the time-series (heteroskedasticity) can bias correlations because periods with higher variance will dominate. In this study such behaviour could originate in our use of climate series composed of both documentary/proxy and instrumental information. We find that this issue is not important in the present study as we have tested this by re-scaling the variance in all series using 21-years running windows, and found only negligible changes in correlations.

### 2.4 The Granger causality test procedure

For those pairs of series showing significant correlations, it is of interest to determine which of two correlated series is the cause of the variations in the other. We know that changes in some climate parameters have a readily detectable effect on grain harvests. Likewise, we also know that changes in harvest output had a substantial effect on the price of grain, but not the other way around (see, e.g., Rahlf, 1996; Persson, 1999; Barquín, 2005; Campbell and Ó Gráda, 2011). Confirming the right direction of causality gives additional indication of the presence of a physical relationship. A formal causality test is the Granger causality test (Granger and Elliott, 1967; Granger, 1969; Reichel et al., 2001). The test considers significant correlations between series that are lagged relative to each other. A statistical causality is indicated if appropriately lagged series correlate while they do not at the opposite lag. While correlation is not causation, the Granger test provides hints for causality between pairs of series because the statistical causation is a necessary, if not sufficient, condition for establishing a physical relationship.

In the procedure two models are written – one for the series in terms of lagged values of itself, and another that additionally includes lagged values of the other series. If the latter model is statistically better than the first one, we see that inclusion of past values of the other series improves the explanatory power of the model with respect to the former simpler model. If we thereafter exchange the roles of the series and test again, we can by inspection understand whether their roles in explaining variance are symmetric or not. Only when the past values of one series helps improve the prediction of another series – and not the other way around – have we found the presence of Granger causality.

Formally, we write

$$y_t = a_0 + a_1 y_{t-1} + a_2 y_{t-2} + \ldots + a_m y_{t-m} + \epsilon \tag{1}$$

$$y_t = a_0 + a_1 y_{t-1} + a_2 y_{t-2} + \ldots + a_m y_{t-m} + b_1 x_{t-1} + b_2 x_{t-2} + \ldots + b_m x_{t-m} + \epsilon \tag{2}$$

$$x_t = a_0 + a_1 x_{t-1} + a_2 x_{t-2} + \ldots + a_m x_{t-m} + \epsilon \tag{3}$$

$$x_t = a_0 + a_1 x_{t-1} + a_2 x_{t-2} + \ldots + a_m x_{t-m} + b_1 y_{t-1} + b_2 y_{t-2} + \ldots + b_m y_{t-m} + \epsilon, \tag{4}$$

where subscripts indicate lags in variables. $\epsilon$ is the residuals, or variance not explained by the model. Equations 1 and 2 are estimated and a Wald hypothesis test (Diggle et al., 2002) is performed to see if Equation 2 is superior to 1. If so, we have preliminary signs of causality from the $x$ series to the $y$ series. We test equations 3 and 4 as well. If Equation 4 does *not* have better explanatory power than Equation 3 then we know that the inclusion of $y$ is unable to improve the explanatory power of $x$ and $y$; i.e., that past values of $x$ help explain present values of $y$ – but not the other way around. The maximum number of lags used, $m$, must be chosen. For this article, we applied the Wald test at the $p = 0.05$ significance level, and allowed lags up to three years ($m = 3$) as longer lags appear unmotivated in the study of climate–harvest (or harvest–price) relationships (see, e.g.,



Bekar, 2019; Huhtamaa et al., 2022; Ljungqvist et al., 2022). The Granger causality is calculated both on the 10-year high-pass filtered data and on linearly detrended data. To evaluate the skill of the Granger causality test in assessing climate–harvest yield relationships, we also calculate Granger causality on harvest yields and grain prices, using a limited price dataset (Table A1), as the true direction of this relationship during this time period is without question (i.e., from harvests to prices).

The study of correlation between series can be confounded by the presence of auto-correlative structure in the series (von Storch and Zwiers, 1999). Series correlation and their significance can be calculated between any level stationary series as long as the presence of autocorrelation in one or both series is taken into account by the inclusion of lagged terms in the model. These are the equation terms at lags $t-1$, $t-2$ et cetera, seen above. We report in Section 3.5 results only of tests that are unambiguous, that is, we consider only Granger causality tests where a clear causality direction is indicated, and do not

consider Granger test results where one or more of the tests involved were not significant.

## 3   Results

### 3.1   Climate–harvest yield relationships in Sweden

The strongest and most consistent climate signal recorded in Swedish grain tithe series is the positive association with summer soil moisture (i.e., wetter conditions = higher yields and *vice versa*) (Fig. 3). This association is found both with 10-year high-

pass filtered and linearly detrended data. Focusing first on the results obtained using high-pass filtered data, the relationship between harvest yields and soil moisture is significant in 57% (phase-scrambling test) and 68% (*t*-test) of the cases for scPDSI and in 68% (phase-scrambling test) and 86% (*t*-test) for SPEI. Notably, all tithe series averaged by type show significant positive correlations with soil moisture (Table A2).

    Stronger correlations than in most county-level tithe series are obtained between the mean of the tithe series and SPEI and

scPDSI with $r = 0.18$ and 0.35 for oats, 0.32 and 0.22 for wheat, 0.39 and 0.30 for barley/rye, and 0.39 and 0.27, respectively, for the average of all the Swedish tithe series. Correlations exceeding $r = 0.40$ against SPEI are found for all grain types in Västmanland County and for oats in Östergötland County. Weak and insignificant correlations are mainly found for the counties of Kopparberg and Värmland located in a wetter and cooler region (Table A2). The relationship with reconstructed precipitation is weak and mainly insignificant in all counties (see further discussion in Section 4.1).

Only a limited number of high-pass filtered Swedish tithe series show a significant, positive or negative, relationship with temperature in any season. There is a minor tendency towards positive correlations between grain yields and Stockholm January–April temperature, though this association is only significant in 11% (phase-scrambling test) and 25% (*t*-test) of the cases. Importantly, this relationship is insignificant with either test for the averaged tithe series. Almost all individual tithe series, and all the averaged tithe series, show a negative association with June–August temperature. Significant negative

correlations are obtained in 21% of the cases with both significance tests using the EuroMed2k reconstruction, while 18% (phase-scrambling test) and 25% (*t*-test) are reached for the Luterbacher et al. (2004) temperature reconstruction. However, importantly, all the averaged tithe series show significant negative correlations with a phase-scrambling based significance test against at least one of the two June–August temperature reconstructions.





The relationship between grain tithes and summer soil moisture weakens when considering linearly detrended. The former
show mean correlations of $r = 0.19$ and $0.12$ with scPDSI and SPEI, respectively (compared to $r = 0.22$ and $0.30$, respectively, for high-pass filtered data; Table A2). Similarly, the SPEI summer hydroclimate reconstruction, showing the strongest association with harvests when using high-filtered data, reveals decreasing correlations when instead using linearly detrended data. On the contrary, the association between reconstructed scPDSI and harvests remains similar (see, further, the discussion in Section 4.4), whereas a stronger scPDSI signal is recorded in the barley/rye average compared to the individual series
(Table A2).

Winter temperature–harvest relationships are weaker, and mainly insignificant, in the linearly detrended data than in the high-pass filtered data. We observe a shift from generally negative summer temperature–harvest correlations (i.e., cold = high yields and *vice versa*) in the high-pass filtered data to weakly positive (i.e., warm = higher yields and *vice versa*), though mainly insignificant, summer temperature–harvest relationships in the linearly detrended data. Using the high-pass filtered data, 21 out
of 28 tithe series reveal a negative summer temperature association, while in the linearly detrended data 20 out of 28 tithe series instead showed a *positive* association – significant or not (Table A2). It is particularly striking that four out of six of the tithe series showing significant negative correlations (phase-scrambling-based test) now instead show positive correlations (significant or not) (Fig. 3).

Despite shorter time-series we also tested the barley and rye tithes separately (Table A3), and found that correlations are, in
general, less significant than for the combined barley/rye tithes. While the overall correlation pattern is similar to the pattern for the combined barley/rye data, some differences in the climate–harvest association can be noted between barley and rye. Rye shows, in general, a stronger positive correlation with summer soil moisture than barley considering high-pass filtered data but not considering linearly detrended data. No significant winter temperature signal is found in barley, Västmanland County being an exception, while rye shows several significant positive correlations, in particular for high-pass filtered data. Furthermore,
we note that several of the Swedish tithe series, including the counties of Stockholm and Uppsala, have gaps and single years of 'outliers' that have a large effect on the results. For example, Uppsala County when considering only the 1690–1740 period using linearly detrended data, we found a positive correlation between summer temperature and barley ($r = 0.39$), rye ($0.44$), barley/rye ($0.24$) (not shown).

### 3.2 Climate–harvest yield relationships in Switzerland

The strongest and most consistent climate signal in harvest yields in Switzerland, when using 10-year high-pass filtered data, is the *negative* relationship between harvests and December–February precipitation (i.e., higher precipitation = lower yields and *vice versa*) (Fig. 4). This association is significant in 70% of the cases considering the phase-scrambling based significance test (in 76% with a *t*-test). The mean correlation between December–February precipitation and the Swiss tithe series is $r = -0.20$, with a range from $-0.06$ (Trachselwald) to $-0.33$ (Gottstatt). Mean correlation for Swiss yield ratio data is $r = -0.12$, with a
range from $0.08$ (St. Gall spelt) to $-0.23$ (Basel rye).

An almost equally strong climate signal, as with winter precipitation, in the high-pass filtered data is the positive association between tithe series and December–February temperature (i.e., higher temperature = higher tithes and *vice versa*). The Pfister





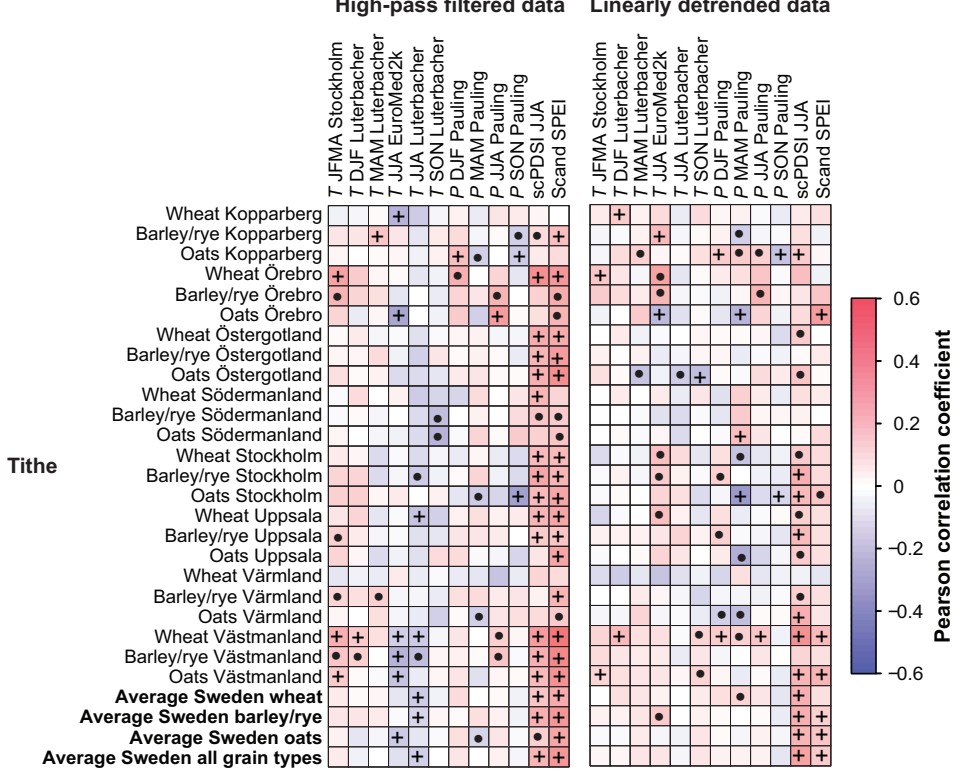

**Figure 3.** Cross-correlation matrix for Sweden between climate series and grain tithe series at a county level for 10-year high-pass Gaussian filtered data (*left*) and linearly detrended data (*right*), respectively. Correlations significant at the $p = 0.05$ level with a $t$-test are marked with a dot ($\cdot$). Values that are additionally significant using the phase-scrambling test are marked with a plus sign (+).

(1992) and Dobrovolný et al. (2010) reconstructions give almost identical results, with 50% significant positive correlations with the phase-scrambling based significance test (63% *vs.* 67% significant with a $t$-test). As opposed to the tithe records, the

yield ratio records mostly show a negative – in some cases significant – correlation with December–February temperature. This negative association is most evident for the Basel rye yield ratio record and the averaged rye yield ratio record (Table A4). The mean correlation between reconstructed Pfister December–February temperature (alternatively the December–February reconstruction by Dobrovolný et al. (2010)), shown in parenthesis) and the Swiss tithe series is $r = 0.18$ (0.19), with a range from $-0.02$ (0.00) (Nidau) to 0.37 (0.40) (Wadenswil). The corresponding correlation for Swiss yield ratio data is a mean of

$-0.06$ ($-0.07$), with a range from 0.08 (0.06) (Zurich spelt) to $-0.22$ ($-0.22$) (Basel rye).

    Many of the Swiss tithe series also reveal a positive correlation with spring and, to a lesser extent, annual mean temperature (Fig. 4). Using the Pfister March–May temperature indices, 27% of the harvest series show significant positive correlations with the phase-scrambling based significance test (33% with a $t$-test), and 30% (45% with a $t$-test) when using the March–May Dobrovolný et al. (2010) temperature reconstruction. Considering annual mean temperature, the Pfister temperature indices show

positive significant correlations to 21% of the harvest series with a phase-scrambling based significance test (27% with a $t$-test).



On the other hand, it also shows two cases of significantly *negative* correlations with a phase-scrambling based significance test (three cases with a *t*-test).

The strongest, and most consistent, negative Swiss harvest–summer temperature association is found for the tree-ring maximum latewood density-based Lötschental summer temperature reconstruction. This association is significant in 36% (phase-scrambling test) and 61% (*t*-test) of the cases. The averaged rye yield ratio for Switzerland shows a significant negative correlation to both temperature and precipitation during most seasons, except for significant positive correlations to summer and autumn precipitation, while the averaged spelt yield ratio only shows a significant negative correlation to winter precipitation and a significant positive one to summer precipitation. In addition, the spelt yield ratio average shows a marginally significant (although only with a *t*-test) negative correlation with reconstructed summer scPDSI and annual groundwater level (Fig. 4; Table A4).

In general, weaker correlations are found using linearly detrended data instead of high-pass filtered data. However, stronger – but mixed – correlations are obtained with the linearly detrended data against reconstructed groundwater levels, scPDSI and SPEI. The mainly positive association between winter temperature and harvests remains, although weakened, while it essentially disappears for spring temperature (Fig. 4). Likewise, the negative relationship between winter precipitation and harvests are retained in linearly detrended data, but considerably weaker and less significant than in high-pass filtered data (Table A4). Furthermore, the already weak positive relationship between summer precipitation and harvests in high-pass filtered data is entirely absent in the linearly detrended data.

Notably, when using linearly detrended data instead of high-pass filtered data, almost all significant correlations between harvests and summer temperature disappears with the phase-scrambling test. Only three of them still remain significant with the less strict *t*-test for the Lötschental summer temperature reconstruction (Fig. 4). Furthermore, while the relationship between Swiss harvests and summer temperature is consistently negative (i.e., colder = high yields and *vice versa*) when using high-pass filtered data, more mixed signals are recorded when using linearly detrended data. Thus, the consistent negative relationship is restricted to the high-frequency domain, and a geographically more varying association prevails when information on longer time-scales are retained (see the discussion in Section 4.4). In the distribution of correlations for Lötschental summer temperatures shown in Fig. 6, we see a distribution of correlations using linearly detrended data symmetric about zero as opposed to the mainly negative clustering of correlations using high-pass filtered data. This implies the presence of an important signal at inter-annual time-scales, and the lack of any meaningful association over longer time-scales. A similar result is seen for Swiss summer precipitation (Fig. 6).

## 3.3 Climate–harvest yield relationships in Spain

No consistent climate–harvest relationship, neither for temperature nor hydroclimate, is found for Spain using either high-passed filtered or linearly detrended data (Fig. 5). The most significant correlations for high-passed filtered are obtained for the March–May season, the main Spanish grain growing period (Section 1), as well as for annual mean values of temperature and precipitation. Several series are negatively correlated with December–February temperatures (Table A5). The direction of the correlations varies between locations (but maintaining local consistency), although mostly negative for temperature





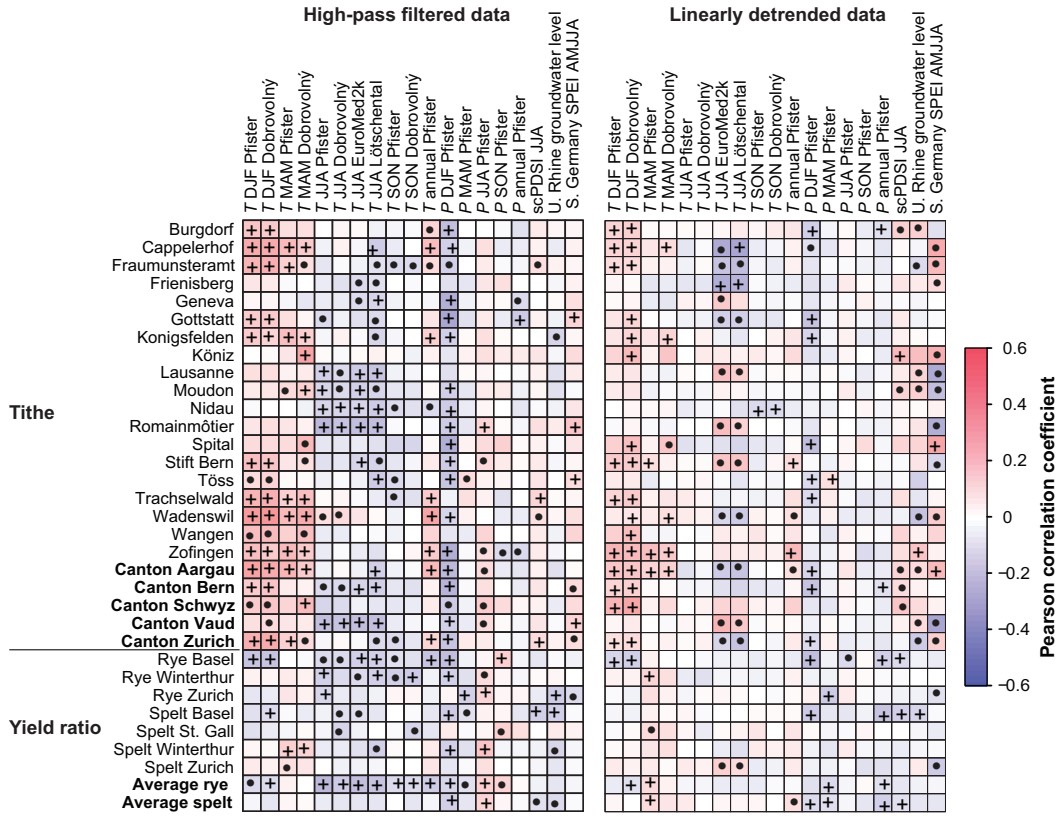

**Figure 4.** Cross-correlation matrix for Switzerland between climate series and grain tithe and yield ratio series for 10-year high-pass Gaussian filtered data (*left*) and linearly detrended data (*right*), respectively. Correlations significant at the $p = 0.05$ level with a *t*-test are marked with a dot (·). Values that are additionally significant using the phase-scrambling test are marked with a plus sign (+).

and positive for precipitation. The tithe series from Aragon and Guadalajara exhibit a different climate signature than most other Spanish series. Non-wheat tithe data from Aragon shows, with a *t*-test, significant positive correlations with September–November temperature as well as with annual mean temperature. Oats tithe data from Guadalajara exhibits, also with a phase-scrambling based significance test, significant positive temperature associations for March–May as well as to the annual mean. The wheat tithe data from Guadalajara correlates significantly positive to June–August temperature with a phase-scrambling

based significance test, whereas the average of the Guadalajara tithe of all grain types is significantly correlated too albeit only with a *t*-test. Furthermore, as opposed to most other Spanish harvest series, all Guadalajara tithe series show a negative – in some cases significantly so – relationships to precipitation during all seasons. All significant correlations, both for high-pass filtered and linearly detrended data, between Spanish yield series for wheat and temperature are negative (i.e., warmer = lower yield and *vice versa*).

Comparatively similar correlation patterns and significance levels, are obtained for Spain when using linearly detrended data – which is in contrast to the findings from Sweden and Switzerland (Sections 3.1– 3.2). In fact, we find no contradicting





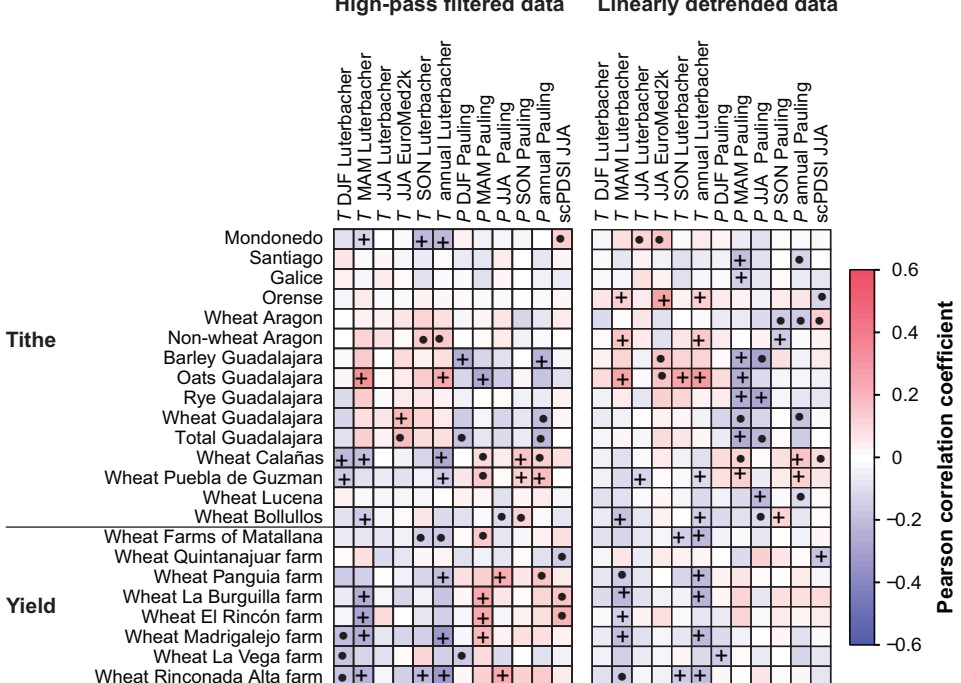

**Figure 5.** Cross-correlation matrix for Spain between climate series and grain tithe and yield ratio series for 10-year high-pass Gaussian filtered data (*left*) and linearly detrended data (*right*), respectively. Correlations significant at the $p = 0.05$ level with a *t*-test are marked with a dot (·). Values that are additionally significant using the phase-scrambling test are marked with a plus sign (+).

significant results between high-pass filtered data and linearly detrended data. However, a more significant negative correlation between the Guadalajara tithe series (covering only the eighteenth century) and March–May precipitation appears when using linearly detrended data (Table A5). Likewise, the positive relationship between March–May precipitation and several of the

yield series disappears. Furthermore, the association with December–February temperature is weakened.

### 3.4   Climate–harvest yield relationships in other regions

Finally, we consider the climate signature of data from regions for which we have only one or a few harvest series (Table 4). Starting with England, high-pass filtered wheat yields show a significant negative correlation to March–May precipitation and barley yields to September–November precipitation (although only significant with a *t*-test). Linearly detrended wheat

data for England exhibit significant positive associations with June–August temperature and negative ones with March–May precipitation (the later only with a *t*-test). The Nuremberg rye tithe record shows a significant negative correlation to spring, summer and autumn temperature (from $r = -0.18$ to $-0.31$) as well as to spring, summer and autumn precipitation and to summer soil moisture (from $r = 0.29$ to $0.38$), when considering high-pass filtered data. However, a negative association is found with December–February precipitation ($r = -0.17$). Fewer of the correlations are significant for the linearly detrended





Nuremberg rye tithe series. An exception is summer soil moisture and summer temperature (the later only with a *t*-test) (Table 4).

The Arles wheat yield ratio record exhibits no significant correlations except for September–November precipitation in high-pass filtered data (and only with a *t*-test). The Alsace tithe data show, using high-pass filtered data, significantly positive correlations with winter temperature and negative ones with summer and autumn temperature. In addition, significantly nega-

tive correlations for winter precipitation are found. Linearly detrended Alsace tithe data shows positive correlations to winter temperature and a disagreement between reconstructions for summer temperature. A significant positive association is found against spring precipitation and a negative one against autumn precipitation. The Beaume tithe record only shows significant negative correlations against winter precipitation (only with a *t*-test) in the high-pass filtered data and negative ones against summer precipitation in the linearly detrended data. In the high-pass filtered French Mediterranean tithe data significant neg-

ative correlations are found against summer soil moisture, whereas a significantly negative correlation is found against spring temperature in the linearly detrended data. The Siena wheat yield record show no significant correlations except a negative one against summer temperature ($r = –0.27$) in linearly detrended data (Table 4).

### 3.5 Granger causality of climate–harvest yield relationships

To establish the direction of causality, in order to reduce the possibility of spurious climate–harvest yield associations, we

performed Granger causality analysis between harvest data and those temperature and hydroclimatic reconstructions with the most significant correlations for Sweden, Switzerland, and Spain (as listed in Tables A2–A5). This means EuroMed2k June–August temperature and Scandinavian SPEI for Sweden, Dobrovolný et al. (2010) December–February temperature and December–February precipitation for Switzerland, and March–May temperature and precipitation for Spain. In addition, we performed a Granger causality analysis between harvest series and regional grain price data (listed in Table A1) for the same

three regions to investigate and preclude the possibility of spurious correlations.

In the case of Sweden, we have 28 harvest (tithe) series to be tested for temperature–harvest relationships of which only a handful provide evidence for a Granger causality relationship. We note that linearly detrended data gives stronger results, especially at lags 1 and 2, than high-pass filtered data. For Switzerland, we have 33 harvest (tithe and yield ratio) series to be tested of which up to two-thirds give usable temperature–harvest results; more than half of these have Granger causality in the

expected direction (i.e., from temperature to yields). Linearly detrended data gives stronger indications of Granger causality at lag 1 and 3, whereas high-pass filtered data gives stronger indications of Granger causality at lag 2. In the case of Spain, where 24 series could have returned Granger causality test results, we find that at most only four do so – therefore, we have no strong evidence for Granger causality temperature–harvest relationships for Spain (Table 5).

Roughly half of the 28 'testable' (i.e., number of pairs that can calculated) pairs from Sweden return results for hydroclimate

(Table A2). Most of these are in the expected direction. Relatively similar results are obtained for high-pass filtered and linearly detrended data. For Switzerland, weak evidence is found for precipitation–harvest relationships with Granger causality. However, linearly detrended data gives us somewhat stronger results at the risk of being spurious. We found for linearly detrended data that eight causalities were significant, and that seven of these were directed from precipitation towards harvests.





**Table 4.** Climate–harvest yield relationships for regions for which we have only one or a few harvest datasets. Bold values indicate correlations significant at the $p = 0.05$ level with an ordinary parametric $t$-test. An asterisk (*) after the correlation value indicates that the correlations also are significant with the more conservative phase-scrambling-based significance test.

| Dataset | $T_{DJF}$ | $T_{MAM}$ | $T_{JJA}$[a] | $T_{JJA}$[b] | $T_{SON}$ | $P_{DJF}$ | $P_{MAM}$ | $P_{JJA}$ | $P_{SON}$ | scPDSI |
|---|---|---|---|---|---|---|---|---|---|---|
| *10-year high-pass filtered data* | | | | | | | | | | |
| Yield England wheat | 0.07 | 0.07 | –0.05 | –0.09 | –0.02 | –0.02 | **–0.19*** | 0.01 | 0.02 | –0.05 |
| Yield England barley | –0.08 | 0.06 | 0.05 | 0.01 | 0.03 | –0.08 | 0.01 | –0.02 | **–0.18** | 0.06 |
| Yield England oats | –0.04 | 0.04 | –0.04 | –0.10 | 0.05 | 0.08 | 0.01 | 0.04 | –0.07 | 0.09 |
| Nuremberg rye tithe | –0.01 | **–0.18** | **–0.25*** | **–0.31*** | **–0.27*** | **–0.17** | **0.29*** | **0.38*** | **0.18*** | **0.34*** |
| Arles wheat yield ratio | 0.12 | –0.09 | –0.01 | 0.06 | 0.06 | –0.05 | 0.13 | –0.03 | **0.19** | –0.01 |
| Alsace tithe | **0.13*** | –0.06 | **–0.15*** | **–0.12** | **–0.13** | **–0.17*** | 0.02 | 0.10 | 0.02 | 0.10 |
| French Mediterranean tithe | –0.10 | –0.12 | 0.09 | 0.04 | 0.08 | –0.03 | 0.10 | 0.10 | 0.03 | **–0.16*** |
| Beaume tithe | 0.07 | 0.00 | –0.09 | –0.10 | 0.13 | **–0.18** | –0.10 | –0.05 | 0.01 | 0.05 |
| Siena yield | –0.10 | –0.08 | 0.07 | 0.08 | 0.05 | –0.15 | 0.00 | –0.06 | 0.08 | –0.14 |
| | | | | | | | | | | |
| *Linearly detrended data* | | | | | | | | | | |
| Yield England wheat | 0.08 | 0.13 | **0.14*** | 0.07 | –0.03 | 0.02 | **–0.13** | 0.01 | –0.05 | –0.04 |
| Yield England barley | –0.06 | 0.03 | 0.07 | 0.10 | 0.06 | –0.01 | 0.07 | 0.02 | –0.05 | –0.01 |
| Yield England oats | –0.02 | 0.06 | –0.01 | 0.11 | 0.04 | –0.01 | –0.04 | 0.06 | –0.05 | 0.09 |
| Nuremberg rye tithe | –0.01 | –0.08 | **–0.31** | –0.08 | 0.02 | –0.08 | –0.07 | 0.06 | 0.00 | **0.16*** |
| Arles wheat yield ratio | 0.13 | –0.11 | –0.03 | –0.04 | 0.04 | –0.07 | 0.06 | –0.03 | 0.12 | –0.03 |
| Alsace tithe | **0.12*** | –0.01 | **0.25** | **–0.11** | 0.01 | –0.05 | **0.24** | –0.08 | **–0.11*** | 0.07 |
| French Mediterranean tithe | –0.08 | **–0.18*** | –0.11 | –0.02 | 0.00 | –0.04 | 0.00 | 0.00 | 0.00 | –0.09 |
| Beaume tithe | 0.09 | –0.08 | –0.07 | –0.04 | 0.11 | –0.09 | 0.06 | **–0.19*** | –0.10 | –0.13 |
| Siena yield | –0.01 | –0.02 | **–0.27** | –0.08 | 0.06 | 0.13 | 0.15 | –0.13 | –0.12 | –0.13 |

[a] Refers to the EuroMed2k June–August temperature reconstruction by Luterbacher et al. (2016) as updated by Ljungqvist et al. (2019).

[b] Refers to the June–August temperature reconstruction by Luterbacher et al. (2004) and Xoplaki et al. (2005).





**Figure 6.** Histograms showing the distribution of key climate–harvest yield correlations. Heavy-coloured bars denote correlations significant also using the conservative phase-scrambling technique and weakly coloured bars denote significant correlations using only the standard parametric *t*-test. The colour scheme follows the matrices in Figs. 3–5.

Only one pointed the other direction. Very few results of precipitation–harvest relationships with Granger causality are obtained
for Spain (Table 5). This is in particular the case using high-pass filtered data.



**Table 5.** Granger causality test results based on tithe and yield series and temperature, hydroclimate, and grain price records. The maximum ($m$) order of the lags tested is three (column 1). Columns 2 to 7 give the results, for 10-year high-pass filtered and linearly detrended data, respectively, in pairs, of the Granger causality test in the notation 'A/B' where 'B' is the number of series that returned unambiguous test results, and 'A' is the number that only allowed for causality in the excepted direction. Wald testing performed at the $p = 0.05$ significance level. After each region name we indicate $n$ the number of tithe and yield series available for testing.

### A Temperature vs. harvests

| | Sweden ($n = 28$) | | Switzerland ($n = 33$) | | Spain ($n = 24$) | |
|---|---|---|---|---|---|---|
| $m$ | High-pass | Linearly detrended | High-pass | Linearly detrended | High-pass | Linearly detrended |
| 1 | 2/5 | 9/12 | 6/10 | 17/20 | 0/0 | 0/1 |
| 2 | 3/3 | 9/9 | 15/23 | 15/16 | 2/3 | 0/1 |
| 3 | 4/4 | 6/7 | 9/11 | 9/14 | 3/4 | 1/2 |

### B Hydroclimate vs. harvests

| | Sweden ($n = 28$) | | Switzerland ($n = 33$) | | Spain ($n = 24$) | |
|---|---|---|---|---|---|---|
| $m$ | High-pass | Linearly detrended | High-pass | Linearly detrended | High-pass | Linearly detrended |
| 1 | 8/14 | 8/10 | 1/3 | 6/6 | 0/0 | 3/4 |
| 2 | 13/16 | 11/15 | 0/2 | 3/5 | 0/0 | 4/4 |
| 3 | 13/17 | 9/10 | 0/2 | 1/4 | 1/1 | 3/3 |

### C Harvests vs. prices

| | Sweden ($n = 28$) | | Switzerland ($n = 33$) | | Spain ($n = 24$) | |
|---|---|---|---|---|---|---|
| $m$ | High-pass | Linearly detrended | High-pass | Linearly detrended | High-pass | Linearly detrended |
| 1 | 20/20 | 14/14 | 15/23 | 7/21 | 14/22 | 10/22 |
| 2 | 19/20 | 15/15 | 13/19 | 10/18 | 10/15 | 10/19 |
| 3 | 18/20 | 15/15 | 9/11 | 8/12 | 7/12 | 12/21 |

For Sweden, for which we have 28 testable harvest series and the Stockholm barley/rye price series, roughly two-thirds return usable results, and in almost all cases the expected direction of Granger causality. The high-pass filtered data gives slightly stronger results. In case of high-pass filtered data, at lag 1, 20 out of 20 unambiguous results are in the expected direction (i.e., from yield to price). The harvest–price relationship for Switzerland, with 33 harvest series and the Basel rye and Zurich spelt price series, return usable results for over half the cases for lag 1 and 2 (Table 5). We tested all the harvest series against both the Basel rye and the Zurich spelt prices. The difference between high-pass filtered and linearly detrended data is small. For Spain, 22 out of 24 series return unambiguous results at lag 1. At lags 2 and 3, almost all return unambiguous results when using linearly detrended data but only about half when using high-pass filtered data. A higher proportion of the results for Spain for high-pass filtered data than linearly detrended data are in the expected direction of Granger causality.




## 4 Discussion

### 4.1 The importance of climate for harvest yields

Relatively weak, but overall regionally consistent, climatic signals have been found in early modern harvest data across Europe (Section 3). The strongest climate signal is the positive association with summer soil moisture in Sweden (Section 3.1), the positive association with winter temperature and negative association with winter precipitation in Switzerland (Section 3.2) and the negative association with spring and annual mean temperature in Spain (Section 3.3). Dispersed harvest data from, e.g., France and Germany echo the negative Swiss association with winter precipitation, the heterogeneous relationship with summer temperature across regions, and that several Swedish regions have a stronger summer drought signal than in any other region (Section 3.4). Interestingly, in Sweden, and partly in Switzerland, the summer temperature signature is frequency-dependent (Section 4.4). The Granger causality test reveals that highly significant climate–harvest associations tend to be in the expected direction (i.e., from climate to yield) (Section 3.5).

Considering auto-correlation (AR1), no systematic differences in the strength of climate–harvest relationships are found between data characterised by high or low AR1 values. The auto-correlative structure of harvest series could be dependent on a number of factors, e.g., informal regulations of local tithes (Leijonhufvud, 2001), inadequate documentation (Le Roy Ladurie and Goy, 1982), access to and shortage of seed grain (Huhtamaa et al., 2022) or possibly through a dependency on a climatic variable with an auto-correlative structure (Esper et al., 2015). In the first three of these hypothesised causes, it could be expected that harvest data with an AR1 higher than that of the climate parameters would have produced weaker correlations. This could in particular affect the hydroclimate signal because precipitation has less memory than temperature (Bunde et al., 2013; Franke et al., 2013; Ljungqvist et al., 2019).

Our results (Section 3.1) obtained using high-pass filtered climate and tithe data from Sweden are, in part, in line with previous studies for later historical periods using first-differenced data (e.g., Edvinsson et al., 2009). However, we do not detect the previously reported winter temperature signal (e.g., Holopainen et al., 2012). The lack of such a signal is in line with results from southern-most Sweden for the eighteenth and nineteenth centuries (Skoglund, 2022). Furthermore, it is interesting to note the appearance of a temperature constraint during the growing season for Örebro and Stockholm counties, which suggests that the boundary between the temperature-constrained cultivation in northern Sweden and the precipitation-constrained cultivation in southern Sweden might historically, during colder climate periods (not shown), have been located further south than previously suggested (e.g., Utterström, 1957; Edvinsson et al., 2009).

For Switzerland, we confirm a significant negative association between winter precipitation and grain harvests (Section 3.2). However, we did not find evidence for a negative association with autumn precipitation as previously reported (Pfister, 1988, 2005, 2007). The winter precipitation signal may, in part, be due to the flushing of nitrogen from manure spread on the fields, which mainly affected the lower part of the Swiss Plateau (Pfister, 1984). We also confirmed the reported positive correlation between warm springs and harvests, but we found the effect of warm winters to be stronger. The stronger winter than spring temperature signature in the Swiss data could partly be related to the circumstance that documentary-based climate reconstructions may better capture cold season variations owing to distinct ice and snow phenology signals. The re-



ported adverse effect of wet summers (Pfister and Wanner, 2021) could not be detected. This could possibly be related to the
fact that we use average data for June to August, while the negative precipitation–harvest association is restricted mainly to
the month of July (Pfister, 1988). Instead, though constrained to the high-frequency domain, we found a negative summer
temperature–harvest relationship indicative of drought sensitivity (Table A5).

The heterogeneous and weak climate–harvest association found for Spain (Section 3.3) is in line with previous results (Bar-
riendos, 2005; Llopis et al., 2020; Moreno et al., 2020). This is presumably directly related to the fact that Spain hosts among
the most heterogeneous climatic conditions in Europe (Rodrigo and Barriendos, 2008). In most of Spain, harvests were gen-
erally dependent on climatic conditions during the winter and spring seasons as they constituted the main grain growing
seasons (Simpson, 1996). In some areas the dry season occurs later and is less intense, as in northern Spain where cereals were
harvested in July, possibly explaining the existence of weak summer climate signals in some instances (Moreno et al., 2020).
On the highland plateau of central Spain, the dry season begins in June and is drier than further north in Spain. In the more
drought-prone southern parts of Spain harvests benefited instead mainly from cooler and wetter winters and springs. In contrast
with most other parts of Europe, in certain parts of Spain – particularly in the south and east – crops were in some instances
irrigated, meaning that the effects of hydroclimatic variability would have been partly mediated through institutional access
to water rights in local irrigation systems (Sarrión, 1995; Palerm-Viqueira, 2010; Catalayud et al., 2022). Furthermore, previ-
ous research has found that there were several outbreaks of crop pests like the fungi common bunt in Spain, reducing wheat
harvests, that were associated with periods of intense rainfall, thus potentially blurring out the drought signal (Moreno et al.,
2020). Moreover, as discussed in Section 4.5, there are limitations to the available hydroclimatic data for the study period. In
addition, research on modern grain yields in Spain have found that different hydroclimate variables (e.g., PDSI, SPEI) vary in
their skill in explaining harvest variations (Peña-Gallardo et al., 2019).

It might appear surprising that Sweden is the study region showing the strongest growing season drought sensitivity on
grain harvest. However, the annual mean precipitation in the main agricultural areas in the southeast of Sweden is only about
half of the rainfall in the main agricultural areas of Switzerland; furthermore, only about 15% of the precipitation falls in
March–May compared to about 25% in Switzerland. Consequently, despite a shorter growing season, Swedish agriculture
is more drought-prone particularly during spring and early summer. Whereas June–August mean temperature is comparable
between Stockholm (17.4°C) and Bern (18.0°C, both from 1991–2020), the Swedish capital receives more sunshine and less
precipitation during those months (789 vs. 696 hours and 189 mm vs. 322 mm). The spring growing season of grain in Spain is
in fact considerably colder compared to the respective summer growing seasons in Sweden and Switzerland. At Guadalajara,
at 700 metre elevation, the March–May mean temperature reaches only 12°C, which is about 5°C colder than the summers at
the Swedish and Swiss sites. Thus, it is not surprising that certain Spanish locations reveal positive correlations of grain harvest
with winter and spring temperatures.

Focusing on certain divergent correlations, we found regarding Swedish tithe data two counties (Kopparberg and Värmland)
without a significant positive summer soil moisture signal (Table A2). In Kopparberg (Dalarna), located more to the north,
temperature seems to be a major limiting climatic factor according to previous research (Edvinsson et al., 2009), whereas
Värmland, located further west, receives more precipitation (Wastenson et al., 1995). Furthermore, the limited number of





significant correlations to soil moisture in Örebro and Södermanland are presumably due to the short period of data coverage
(1665–1720) for the former and the limited length (1572–1681) combined with large data gaps in the latter (Leijonhufvud,
2001; Hallberg et al., 2016). For Switzerland, the few harvest series with an insignificant negative association between winter
precipitation and harvest are all from mountainous locations. Here, topography prevents the collection of stagnant water in the
fields at the same time as excessive winter precipitation comes generally as snow rather than rain. Furthermore, such locations
had a strong focus on pastoral farming, resulting in a better availability of manure to fertilise the grain fields, making winter
rain washout a less critical problem (Pfister, 1975, 1995; Pfister and Wanner, 2021).

## 4.2 Climate–grain harvest *versus* climate–grain price relationships

Tithe and yield data represent local to regional harvest yields where climatic influences are masked by local-scale stochastic
weather influences. Grain prices, on the other hand, are influenced by production averages across larger regions. Thus, grain
prices can be expected to be more driven by large-scale climatic conditions (Ljungqvist et al., 2022), in particular of temperature
that co-varies over large regions as opposed to precipitation (Büntgen et al., 2010; Ljungqvist et al., 2016). Therefore, grain
prices show stronger associations to temperature than harvest data. Up to 40% of the decadal-scale mean European grain price
variability can be explained by changes in summer temperatures alone when the low-frequency variations are retained in both
the climate and price data (Ljungqvist et al., 2022). On the contrary, the association with hydroclimate, and precipitation in
particular, can be expected to decrease rather than increase with larger spatial scales (represented by grain prices) owing to its
spatially heterogeneous character.

   Relatively similar correlation patterns are obtained between climate and harvest data for most grain types under investigation
(Section 3). Thus, the significant and negative temperature–grain price association (Esper et al., 2017; Ljungqvist et al., 2022),
in contrast to the weaker and partly opposite temperature association with harvest data found here, cannot be explained by
differences in grain types or by the co-influence of wheat on the prices of other grain types (for the latter, see Ljungqvist et al.,
2022). One possible contributing factor why grain prices were significantly lower when temperatures were higher, and *vice
versa*, could be related to drying and storage of grain. Colder growing seasons, tending to be wetter, meant increased risks of
mould and fungi also after the harvest (Pfister, 2005). The harvested grain in wetter and cooler conditions, as compared to in
drier and warmer conditions, could also be of an inferior quality (Pfister and Wanner, 2021). Inferior grain quality could cause
a lower market price without necessary meaning an unusually low total harvest quantity or yield ratio.

## 4.3 Climate–grain harvest associations during years of harvest failures

The comparatively weak correlations between seasonal climate parameters and grain harvest yields should not be interpreted
as that climate conditions played a minor role for harvest failures and subsequent food crises in early modern Europe. While
seasonal climate conditions only explain a fraction of the yield variations during most years, the majority of the severe har-
vest failures – those that could trigger famines during the early modern period – occurred during years with cold springs
and summers, with prolonged periods of excessive precipitation right before, or during, harvest time (Pfister and Wanner,



2021). Especially, consecutive years of poor harvests were associated with cold climatic anomalies (Ljungqvist et al., 2021). Furthermore, excessive precipitation and high relative humidity favoured plant plague development (Moreno et al., 2020).

A linear relationship between climate and harvests can only be expected within certain ranges (e.g., Yin et al., 1995). For most crops and locations, both excessive cold/heat and wetness/dryness would have been detrimental and, hence, the climate–
harvest correlation is obscured by years with contradictory effects. For example, in Spain drought could negatively affect wheat harvests, but excessive wetness could likewise reduce harvests through facilitating the development of crop pests like the common bunt. In contemporary studies on harvest yields it has been clearly demonstrated that a consideration of the full range of impacts of a given climate indicator exhibit non-linear, roughly bell-shaped, relationships (Dell et al., 2014). This obviously puts limitations to linear estimations of the harvest–climate relationships (as conducted here).

Climate played a far larger role for 'shocks' to the food supply than what the relatively low climate–harvest correlations obtained here would imply. Very adverse climatic conditions for grain agriculture were responsible, partly or entirely, for most major food crisis in early modern Europe (Alfani and Ó Gráda, 2017, 2018). This was especially the case as the same climatic conditions that lowered grain yields – cold springs and wet summers or severe droughts – also decreased pasture productivity and fodder availability (Pfister, 1988; White et al., 2018; Brázdil et al., 2019; Huhtamaa and Ljungqvist, 2021); it favoured
the development and spread of epizootic diseases among domestic animals (White, 2014; Newfield, 2015). Excess mortality among domestic animals, in turn, decreased the availability of manure to fertilise the grain fields, thus, lowering the yields in subsequent years (Pfister and Wanner, 2021).

## 4.4   High- and low-frequency climate–harvest relationships

The high-pass filtered data, only containing high-frequency information, and the linearly detrended data, containing low-
frequency information too, have partly revealed different correlation patterns (Sections 3.1– 3.2). In the case of summer temperature for Sweden and Switzerland, the high-pass filtered and the linearly detrended data even show *opposite* directions of correlations (Fig. 6). Non-stationary trend shifts could be related to, for example, a more frequent prevalence of adverse climate or weather conditions for grain crops during colder climate periods, delayed onsets of growing seasons, frost events during the growing season, and excessive precipitation at harvest time occurring more frequently when the climate was colder (Huhtamaa
and Ljungqvist, 2021; Pfister and Wanner, 2021). This indirect relationship between poorer (better) yields and colder (warmer) climate, however, disappears when using high-pass filtered data, possibly because the low-frequency variance is expected to predominantly contain a temperature signal (Fig. 6).

The Swedish summer temperature–grain harvest association shows the strongest change with frequency (Table A2). The negative association with summer temperature disappears, and the summer temperature association even turns positive (Sec-
tion 3.1). However, when considering barley and rye separately the results are somewhat more varied (Table A3). Using linearly detrended data, the results were more sensitive to the cut-off period as well as gaps and outliers in the Swedish tithe data in the sixteenth and early seventeenth centuries. To better constrain early modern climate–harvest relationships across all frequency bands, we would need to acquire harvest data that adequately capture multi-decadal grain yield variability and that are not dominated by trends on these time-scales unrelated to actual growth conditions. The frequency-opposed trends, and overall





low correlation values, between even nearby tithe and yield ratio records – even after removing linear trends – demonstrate that such data are in short supply for the early modern period.

## 4.5 Limitations related to the climate data

There are more uncertainties and sources of biases in the low-frequency than in the high-frequency domain for all utilised data: palaeoclimate, tithe and yield data. This likely explains the generally stronger, and more significant, correlations found in
high-pass filtered data compared to linearly detrended data. Considering different drought indices, for Sweden, SPEI signals appear to be limited to mainly high-frequency variability compared to scPDSI (Table 3). Likely, the scPDSI reconstruction overestimates the low-frequency soil moisture signal and contains a substantial temperature signal (Seftigen et al., 2017). As the temperature spectrum contains more low-frequency information compared to precipitation (Bunde et al., 2013), the reconstructed soil moisture signal might, at long time-scales, be dominated by temperature-driven evapotranspiration processes
rather than precipitation (Baek et al., 2017; Ljungqvist et al., 2019).

The weak correlations for the seasonally resolved Luterbacher et al. (2004) temperature and Pauling et al. (2006) precipitation reconstructions, compared to the Pfister (1992) temperature indices and the Dobrovolný et al. (2010) temperature reconstruction from Switzerland, could also be affected by decreasing quality in these gridded reconstructions back in time. Prior to the late seventeenth century, the Luterbacher et al. (2004) temperature and Pauling et al. (2006) precipitation recon-
structions are solely based on proxy data which only explain a fraction of the actual seasonal climate variability. For example, both the Luterbacher et al. (2004) temperature field reconstruction and the Pauling et al. (2006) precipitation field reconstruction is lacking local input data from Sweden during the period covered by the Swedish tithe series – thus, the lack of correlation with the Swedish tithe data comes as no surprise. Furthermore, the climate models used to derive field reconstructions from the sparse data coverage in these two field reconstructions are very out-of-date. Nevertheless, the stronger seasonal correlations
of the Luterbacher et al. (2004) and Pauling et al. (2006) reconstructions with those Spanish tithe and yield records that only cover the eighteenth century indicate greater skill with the inclusion of early instrumental records in these reconstructions. This likely implies that any lack of significant correlations between seasonal precipitation and harvest data is due to limitations of proxy-based precipitation reconstructions. Our results highlight the need to develop new regional precipitation reconstructions, and improve the quality of existing ones, or precipitation field reconstructions with a denser and longer proxy data network.
The documentary-based Pfister (1992) indices of temperature and precipitation and the Dobrovolný et al. (2010) temperature reconstruction, used for Switzerland, result in numerous significant correlations. However, they may contain limitations in the low-frequency domain as documentary-based climate reconstructions based on the so-called index method typically are unable to capture the full amplitude of low-frequency climate variations (Adamson et al., 2022). This may partly explain the high and significant correlations for Switzerland when using high-pass filtered data compare to linearly detrended data (see Fig. 4;
Table A4). It should, furthermore, be noted that for certain seasons and periods the Pfister (1992) and Dobrovolný et al. (2010) reconstructions include some harvest-related proxy data, thus these climate reconstructions are not fully independent from the harvest yield series they are correlated against.



## 4.6 Limitations related to the harvest data

Possible limitations in the tithe and yield data could weaken climate–harvest relationships. For example, the Swedish tithe data
contain numerous gaps, and also years when data may be considered uncertain (Myrdal and Söderberg, 1991). Furthermore,
the Swedish tithe data for different counties partly cover different periods (Leijonhufvud, 2001; Hallberg et al., 2016). These
periods had somewhat different prevailing climate conditions (Luterbacher et al., 2004; Xoplaki et al., 2005; Leijonhufvud
et al., 2010; Cook et al., 2015; Seftigen et al., 2017; Ljungqvist et al., 2019). The main uncertainty regarding Swedish tithe data
is the unknown variation in the number of farms paying tithes (equalling gaps in the sources) and the fact that the proportion of
tithes to actual harvest may decreased over time (Leijonhufvud, 2001). Over the decades, the practices may have become more
standardised as Hegardt (1975) showed it to be the case for the tithes collected for Uppsala University. In some areas, the tithes
were fixed at an average amount (Le Roy Ladurie and Goy, 1982), but those data are not included in this study. For Switzerland,
an uncertainty in estimating the actual harvest size is related to the fact that tithes were sold at auctions rather than collected
in the field, although the authorities' controls were rather strict (Pfister, 1979). The Spanish tithe data are mainly derived from
rented land and are not fully representing the annual grain production (Llopis et al., 2018) or representing a mixture of tithes
paid in grain and in money (Kain, 1979). A problem in detecting climatic signals in grain harvests using tithe data is that tithes
frequently consist of a mixture of different types of grains with exact proportions of each changing over time (Le Roy Ladurie
and Goy, 1982).

The varying climate sensitivity, between tithe records and yield ratio records from the same region, is not surprising as soil
properties, elevation, and micro-climatic conditions additionally affected the climate–harvest yield associations. Variations in
the yield data may partly be related to changes in farming methods and/or availability of manure for fertilisation (Pfister and
Wanner, 2021). Yield ratios of an individual farm may not be representative even for the surrounding region. An example for
such a bias is the Basel rye yield ratio series (Head-Köenig, 1979) that has a significant positive correlation to winter tem-
perature when almost all other Swiss series show a negative, mostly significant, relationship to winter temperature (Fig. 4;
Table A4). This peculiarity probably reflects local growth conditions and agricultural practices. The individual yield series
can also contain different climate signals because the individual farmers had great possibilities to alter their own productiv-
ity through agricultural practises such as differences in fertilisers, weeding, skills in ploughing, sowing intensity (per area)
as well as different possibilities and opportunities to financial access that would allow for higher-quality agricultural inputs
(better, heavier ploughs, stronger animals and so forth) (Simpson, 1996; Hoffman, 1996; Gadd, 2011; Myrdal, 2011; Martínez-
González et al., 2020). Such differences may to some extent off-set potential climate signals at the local (farm) level, but these
effects will be mitigated when averaging spatially (and in averaged tithes series).

The lack of non-ambiguous trends in many Swedish tithe series and some Swiss tithe and yield ratio series during the
study period implies a sensitivity to non-linear structural breaks.These structural breaks could be related to climatic variability,
but also to non-climatic factors, e.g. (informal) changes to tax collecting (Leijonhufvud, 2001) or changes in farming prac-
tices (Myrdal, 2011). Thus, it is reasonable to conclude that a much stronger climate forcing on the harvests in fact prevailed
than the one detected here.





### 4.7 Human agency dominating long-term harvest yield changes

The relatively weak association between climate variations and harvest yields stem, besides data-related issues in both the harvest and climate data (see above), from a number of factors including: (a) fading and non-linear relationships between seasonal climate and harvest yields at locations far from the climatological limits of a certain crop; (b) the fact that poor harvests often resulted from single days or weeks of unfavourable weather conditions rather than seasonal average climate conditions; (c) the large degree to which harvest yields depended on factors related to human agency (e.g., socio-political and socio-economic setting, infrastructure and technology). The relatively weak association between climate and harvest yield variations emphasises the importance of not exaggerating the role of climate at the expense of other factors even when studying climate-dependent activities like agriculture (*sensu* Hulme, 2011; Haldon et al., 2018; van Bavel et al., 2019).

Early modern harvests were, beside climatic factors, influenced by pests and diseases (partly climate-related), seed quantity and quality, manure availability, access to draught animals, availability of labour force, armed conflicts, workforce health, and market conditions (Simpson, 1996; Gadd, 2011; Myrdal, 2011; Moreno et al., 2020; Pfister and Wanner, 2021). In early modern Europe, fertilisers and labour were limited resources that affected the intensity and efficiency with which land could be cultivated (Pfister and Wanner, 2021), and therefore also grain production. The lack of seed grain after consecutive harvest failures could reduce the harvest also in subsequent year(s) of good growth conditions (Appleby, 1979; Pfister, 1990; Bekar, 2019). This has, for example, also been demonstrated to have been a major problem in early modern Finland (Huhtamaa et al., 2022), but even in non-peripheral rich agricultural districts as Scania (Brasch, 2016). Grain storage and trade eventually reduced or eliminated the effect of seed shortages after harvest failures (Krämer, 2015).

The genetic variety of early modern grain types was larger than today and included a wide diversity of varieties both locally and regionally. For example, in Sweden there existed varieties of extremely fast-growing spring barley that could be sown in late spring (Leino, 2017). Such a genetic diversity of grain varieties, adapted to local environmental conditions (Forsberg et al., 2015, 2019), complicates comparisons between climate–harvest relationships throughout time as well as between regions. It is possible, but generally unknown, whether the same genetic variety was grown at a certain location throughout the entire early modern period. Farming practices, such as sowing time and crop rotation, might have altered with climatic changes as an adaptive measure. However, during colder periods, characterised by shorter growing seasons, the length of the agricultural season (i.e., most of the agricultural work was performed) was reduced.

### 5 Conclusions and outlook

Using a coherent framework of statistical analyses, we re-assessed the temperature and hydroclimate signatures in grain harvest data among diverse environmental settings of early modern (*c.* 1500–1800) Europe. We detected mainly regionally consistent, but overall relatively weak, climate–harvest relationships. Even though the majority of the severe food crises during the early modern period were connected to climate anomalies, only a fraction of documented grain harvest variations can be explained by the available climate data. The extent of historical climate–harvest associations revealed here is similar to contemporary climate–grain yield relationships in Europe. Moderately strong and consistent positive relationships are found between summer



soil moisture and harvests in Sweden, and between winter temperature and harvests in Switzerland. In addition, negative relationships are found between winter precipitation and Swiss harvests. Spatially heterogeneous, and mostly weak, climate–harvest relationships are found in Spain. The limited strength of the climate–harvest associations are contrasting with the comparatively strong climate–grain price relationships previously reported by Esper et al. (2017) and Ljungqvist et al. (2022) among others. We also found that climate–harvest correlations can change direction with frequency and in some regions with season. This is particularly the case for the association between summer temperature and harvest in Sweden and, to a lesser extent, in Switzerland. In these instances, a negative relationship prevailed in the high-frequency domain, in contrast to a positive relationship when information over longer time-scales is retained in the harvest and climate data.

Broader empirical and methodological implications of this study include: (1) the need to develop precipitation reconstructions for the early modern period as the arguably greatest limitation to constrain climate–harvest relationships arise from the lack of such data; (2) continued efforts to digitise already published harvest data and extract new yield series from original archival research for additional regions and to reduce data gaps and uncertainties; (3) explore changes in the frequency and the causality between climate and weather extremes and their effects of harvest yields; (4) further assess potential differences between the climate signal embedded in harvest yield and grain price data including response differences to summer temperature. Related to this problem is the fact that the harvest data let us assess grain yield quantity but not quality which also affects price. Further studies are needed to evaluate the relationships between climate, weather, and the quantity and quality of yields. A next step could include the application of such methods as multivariate singular spectrum analysis of combined datasets on climate, grain yields, and grain price to extract consistent patterns from collections of data.

*Code availability.* We have used **IDL** and **R** (R Core Team, 2022) version 3.6.3 to program the analysis codes used in this work. In **R** we used the 'aod' package (Lesnoff et al., 2012) for the Wald test, and the package 'corrplot' (Wei and Simko, 2021) to generate the correlation matrices. Data were read from mixed files using 'base' libraries for text files or 'openxlsx' (Schauberger and Walker, 2021) for Excel spreadsheet files. The maps in Fig. 1 were drawn in ArcMap 10.6 (program package of ArcGIS, ESRI (2017)) and edited in Adobe Illustrator.

*Data availability.* The majority of the tithe and yield data can be found in tables in the cited publications listed in Tables 1– 2. The Swedish tithe data until 1680 is available from tables published in Leijonhufvud (2001) and the Swedish tithe data after 1665 are described and published in Hallberg et al. (2016) and available at: https://snd.gu.se/sv/catalogue/study/snd0996. All the climate data are digitally available from the National Oceanic and NOAA Paleoclimatology/World Data Center for Paleoclimatology: https://www.ncei.noaa.gov/products/paleoclimatology. Most grain price series listed in Table A1 are included in the Allen-Unger Global Commodity Prices Database (Allen and Unger, 2019) available at: http://www.gcpdb.info/data.html. The Stockholm grain prices series from Edvinsson and Söderberg (2010) is available from Historical Monetary Statistics of Sweden at Sweden's central bank (*Sveriges Riksbank*) at: https://www.riksbank.se/en-gb/statistics/historical-monetary-statistics-of-sweden/



*Author contributions.* F.C.L. designed the study, collected the data, performed exploratory data analysis, interpreted the results and did most of writing. B.C. detrended the data, calculated the correlations and their significance, contributed to the interpretation of the results, drafted method section text and conducted exploratory data analysis. J.E. provided input on article structure, suggested references and edited the text. H.H. provided input on the original study design, contributed to the interpretation and gave critical feedback on the text. L.L. provided the Swedish tithe data, prepared it for this study and drafted part of the text about Swedish tithe data and its limitations. C.P. drafted text about the Swiss tithe and yield ratio data, provided literature references and contributed to the interpretation of the results. A.S. contributed to the interpretation of the results, draw the maps in Fig. 1 in ArcMap 10.6 and Adobe Illustrator and provided input on article structure and edited the text. M.K.S. drafted the current state of research regarding Swedish climate–harvest relationships, contributed with text about Spanish agriculture, provided literature suggestions, gave input on detrending methods, conducted exploratory data analysis, and edited the text. P.T. suggested and calculated the Granger causality analysis, produced the correlation matrices, curated the data collection, contributed to the interpretation of the results and drafted most of the method section.

*Competing interests.* The authors have declared no conflicts of interest for this article.

*Acknowledgements.* F.C.L. acknowledges a Visiting Researcher stay at the Institute of History, University of Bern, that allowed him time to work on this article. Dr. Maria Wallenberg Bondesson helped with digitising data from publications and translating metadata from French. We thank Prof. Enrique Llopis, Universidad Complutense de Madrid, for sharing the Spanish yield data published in Llopis et al. (2020), and Prof. Rafael Barquín, Universidad Nacional de Educación a Distancia, for sharing yield data published in Barquín (2005). This article is dedicated to the memory of Prof. em. Johan Söderberg (1950–2022), Department of Economic History and International Relations, Stockholm University.

*Financial support.* F.C.L. and A.S. were supported by the Swedish Research Council (Vetenskapsrådet, grant no. 2018-01272), B.C. by the NordForsk-funded Nordic Centre of Excellence project (grant no. 76654) Arctic Climate Predictions: Pathways to Resilient, Sustainable Societies (ARCPATH), H.H. by the Swiss National Science Foundation (grant no. PZ00P1_201953) and the Swiss State Secretariat for Education, Research and Innovation funded European Research Council project (ERC contract MB22.00030), P.T. and B.C. received funding from the Danish State through the National Centre for Climate Research (NCKF), and J.E. by the Ministry of Education, Youth and Sports of the Czech Republic for the project SustES ($CZ.02.1.01/0.0/0.0/16_019/0000797$), and the European Research Council (ERC-2019-AdG grant no. 882727). F.C.L. conducted the work with this article as a Pro Futura Scientia XIII Fellow funded by the Swedish Collegium for Advanced Study through Riksbankens Jubileumsfond.





## Appendix A

**Table A1.** Grain price data used for Granger causality analysis. Information of grain type, covered period, missing values ('gaps' in percentage), the auto-correlation coefficient for lag 1 year, AR1, and the data source(s).

| Dataset | Type | Period | Gaps | AR1 | Source |
|---|---|---|---|---|---|
| Barcelona | Wheat | 1501–1800 | 3.01% | 0.83 | Feliu (1991) |
| Basle | Rye | 1501–1797 | 23.31% | 0.82 | Hanauer (1878) |
| Madrid | Wheat | 1501–1799 | 8.67% | 0.77 | Hamilton (1934, 1947) |
| New Castile | Barley | 1504–1750 | 15.85% | 0.73 | Hamilton (1934, 1947) |
| Stockholm[a] | Barley/rye | 1500–1800 | 0.30 | 0.83 | Edvinsson and Söderberg (2010) |
| Valencia | Wheat | 1500–1789 | 0.67% | 0.90 | Hamilton (1936) |
| Zürich | Spelt | 1540–1877 | — | 0.59 | Studer (2015) |

[a] An equal share of barley and rye prices whenever both are available.





**Table A2.** Climate–harvest correlations for Sweden. Results using high-pass filtered data shown to the left of 'l' and results using linearly detrended data to the right of 'l'. Bold values indicate correlations significant at the $p = 0.05$ level with a $t$-test and an asterisk (*) significant correlations with the phase-scrambling test. All correlations for Sweden, and their significance, are shown in Fig. 3.

| Location | $T_{JFMA}$ | $T_{JJA}$ | scPDSI | SPEI |
|---|---|---|---|---|
| *Barley/rye* | | | | |
| Kopparberg County | 0.10 l –0.03 | 0.09 l **0.26**\* | **0.15** l 0.11 | **0.24**\* l –0.07 |
| Örebro County | **0.20** l 0.20 | –0.10 l **0.28** | 0.17 l 0.11 | **0.32** l 0.23 |
| Östergötland County | 0.05 l 0.05 | –0.06 l –0.08 | **0.28**\* l 0.02 | **0.34**\* l 0.07 |
| Södermanland County | –0.01 l 0.01 | 0.00 l –0.12 | **0.21** l 0.06 | **0.20** l 0.00 |
| Stockholm County | 0.16 l –0.01 | –0.04 l **0.20** | **0.34**\* l **0.29**\* | **0.28**\* l 0.13 |
| Uppsala County | **0.18** l 0.07 | –0.03 l 0.11 | **0.21**\* l **0.25**\* | **0.22**\* l 0.12 |
| Värmland County | **0.16** l 0.00 | –0.05 l 0.04 | 0.06 l **0.17** | **0.27**\* l 0.12 |
| Västmanland County | **0.22** l 0.09 | **–0.28**\* l 0.11 | **0.30**\* l 0.16 | **0.45**\* l 0.12 |
| *Oats* | | | | |
| Kopparberg County | 0.03 l –0.04 | 0.04 l 0.05 | 0.07 l **0.19**\* | 0.15 l 0.03 |
| Örebro County | 0.15 l –0.06 | **–0.34**\* l **–0.25**\* | 0.13 l 0.14 | **0.27** l **0.39**\* |
| Östergötland County | 0.12 l 0.12 | –0.13 l –0.06 | **0.30**\* l **0.22** | **0.44**\* l 0.03 |
| Södermanland County | 0.04 l –0.01 | –0.07 l 0.04 | 0.17 l 0.02 | **0.22** l 0.14 |
| Stockholm County | 0.17 l 0.02 | 0.08 l 0.16 | **0.25**\* l **0.26**\* | **0.34**\* l **0.21** |
| Uppsala County | 0.16 l –0.03 | –0.07 l 0.15 | 0.10 l **0.20** | **0.33**\* l 0.11 |
| Värmland County | 0.04 l –0.04 | –0.04 l –0.03 | 0.14 l **0.31**\* | **0.20** l 0.09 |
| Västmanland County | **0.20**\* l **0.16**\* | **–0.22**\* l 0.10 | **0.25**\* l **0.28**\* | **0.42**\* l **0.28**\* |
| *Wheat* | | | | |
| Kopparberg County | –0.06 l 0.08 | **–0.27**\* l 0.13 | 0.04 l 0.09 | 0.00 l 0.12 |
| Örebro County | **0.36**\* l **0.22**\* | 0.02 l **0.35** | **0.44**\* l 0.24 | **0.39**\* l –0.06 |
| Östergötland County | 0.02 l 0.00 | –0.02 l –0.01 | **0.32**\* l **0.20** | **0.31**\* l 0.00 |
| Södermanland County | –0.02 l –0.01 | 0.06 l 0.09 | **0.27**\* l 0.12 | 0.15 l 0.07 |
| Stockholm County | 0.08 l –0.15 | –0.02 l **0.24** | **0.26**\* l **0.20** | **0.26**\* l 0.13 |
| Uppsala County | 0.10 l 0.07 | –0.02 l **0.11** | **0.28**\* l **0.25** | **0.33**\* l 0.12 |
| Värmland County | –0.11 l –0.14 | 0.08 l –0.22 | 0.17 l –0.06 | 0.15 l –0.10 |
| Västmanland County | **0.29**\* l 0.15 | **–0.18**\* l 0.13 | **0.37**\* l **0.42**\* | **0.51**\* l **0.25**\* |
| | | | | |
| *Average wheat tithe* | 0.07 l 0.00 | –0.10 l 0.07 | **0.22**\* l **0.30**\* | **0.32**\* l 0.12 |
| *Average oats tithe* | 0.06 l –0.02 | **–0.20**\* l –0.04 | 0.18 l **0.28**\* | **0.35**\* l **0.22**\* |
| *Average barley/rye tithe* | 0.10 l 0.06 | –0.03 l **0.19** | **0.30**\* l **0.30**\* | **0.39**\* l **0.17**\* |
| *Average all tithe* | 0.07 l 0.04 | –0.12 l 0.09 | **0.27**\* l **0.36**\* | **0.39**\* l **0.20**\* |




**Table A3.** Climate–harvest correlations for Sweden for barley and rye tithe data separately. Results using high-pass filtered data shown to the left of 'I' and results using linearly detrended data to the right of 'I'. Bold values indicate correlations significant at the $p = 0.05$ level with a $t$-test and an asterisk (*) significant correlations with the phase-scrambling test.

| Location | $T_{JFMA}$ | $T_{JJA}$ | scPDSI | SPEI |
|---|---|---|---|---|
| *Barley* | | | | |
| Kopparberg County | 0.00 I –0.08 | 0.15 I **0.26**\* | 0.10 I 0.06 | **0.16** I –0.01 |
| Örebro County | 0.07 I 0.19 | –0.05 I 0.28 | –0.10 I –0.11 | 0.15 I 0.07 |
| Östergötland County | 0.00 I –0.07 | **–0.20** I –0.08 | **0.10** I –0.03 | **0.28**\* I 0.05 |
| Södermanland County | –0.01 I 0.10 | 0.00 I 0.06 | **0.21** I 0.01 | 0.23 I 0.01 |
| Stockholm County | 0.02 I 0.02 | –0.01 I 0.12 | **0.21** I **0.24**\* | 0.16 I 0.09 |
| Uppsala County | 0.14 I –0.02 | –0.04 I 0.11 | **0.21** I **0.25** | **0.29**\* I 0.13 |
| Värmland County | 0.19 I 0.24 | 0.03 I 0.04 | **0.34**\* I **0.34**\* | **0.27**\* I 0.25 |
| Västmanland County | **0.29**\* I **0.24**\* | **–0.28**\* I –0.02 | **0.40**\* I **0.30**\* | **0.44**\* I **0.26**\* |
| *Rye* | | | | |
| Kopparberg County | **0.19** I 0.03 | 0.10 I **0.28**\* | **0.24**\* I 0.06 | **0.45**\* I –0.06 |
| Örebro County | **0.30** I 0.23 | 0.13 I 0.13 | 0.11 I –0.01 | **0.30** I –0.18 |
| Östergötland County | 0.12 I 0.08 | –0.13 I –0.08 | **0.39**\* I 0.09 | **0.51**\* I 0.04 |
| Södermanland County | –0.02 I 0.03 | –0.07 I –0.11 | 0.20 I –0.03 | 0.22 I –0.03 |
| Stockholm County | –0.04 I –0.10 | 0.08 I 0.05 | **0.27**\* I 0.20 | 0.15 I 0.03 |
| Uppsala County | 0.10 I –0.05 | –0.07 I 0.01 | **0.21** I **0.27** | **0.27** I 0.13 |
| Värmland County | **0.21** I **0.19**\* | –0.04 I **0.27**\* | 0.00 I 0.09 | **0.36**\* I 0.06 |
| Västmanland County | **0.29**\* I **0.16** | **–0.22**\* I –0.07 | **0.46**\* I **0.31**\* | **0.40**\* I 0.12 |





**Table A4.** Climate–harvest correlations for Switzerland. Results using high-pass filtered data shown to the left of ' | ' and results using linearly detrended data to the right. Bold values indicate correlations significant at the $p = 0.05$ level with a $t$-test and an asterisk (*) significant correlations with the phase-scrambling test. $T_{DJF}$ = Dobrovolný et al. (2010) DJF temperature reconstruction; $T_{JJA}$ = Büntgen et al. (2006) Lötschental JJA temperature reconstruction; $P_{DJF}$ = Pfister (1992) DJF precipitation indices; $P_{JJA}$ = Pfister (1992) JJA precipitation indices. All correlations for Switzerland, and their significance, are shown in Fig. 4.

| Dataset | $T_{DJF}$ | $T_{JJA}$ | $P_{DJF}$ | $P_{JJA}$ |
|---|---|---|---|---|
| Burgdorf | **0.20**\* \| **0.17**\* | –0.06 \| 0.09 | **–0.24**\* \| **–0.18**\* | 0.00 \| 0.03 |
| Cappelerhof | **0.31**\* \| **0.16**\* | **–0.20**\* \| **–0.35**\* | **–0.18**\* \| **–0.13** | 0.12 \| 0.04 |
| Fraumünsteramt | **0.29**\* \| **0.13**\* | **–0.13** \| **–0.24** | **–0.18** \| –0.10 | 0.09 \| –0.02 |
| Frienisberg | 0.08 \| 0.03 | **–0.12** \| **–0.27**\* | –0.11 \| –0.12 | 0.10 \| 0.09 |
| Geneva | 0.05 \| 0.04 | **–0.16**\* \| 0.12 | **–0.29**\* \| –0.09 | 0.03 \| –0.03 |
| Gottstatt | **0.20**\* \| **0.15**\* | **–0.13** \| **–0.18** | **–0.33**\* \| **–0.22**\* | 0.08 \| 0.07 |
| Königsfelden | **0.17**\* \| **0.16**\* | **–0.15** \| –0.09 | **–0.18**\* \| **–0.15**\* | 0.12 \| 0.05 |
| Köniz | 0.01 \| **0.26**\* | 0.06 \| 0.07 | –0.09 \| –0.11 | 0.12 \| 0.03 |
| Lausanne | 0.12 \| 0.09 | **–0.22**\* \| **0.18** | –0.11 \| –0.04 | 0.05 \| –0.01 |
| Moudon | 0.10 \| 0.10 | **–0.15** \| 0.05 | **–0.19**\* \| –0.04 | 0.09 \| 0.08 |
| Nidau | 0.00 \| 0.03 | **–0.20** \| –0.12 | **–0.20**\* \| –0.06 | 0.05 \| –0.01 |
| Romainmôtier | 0.12 \| 0.04 | **–0.24** \| 0.15 | **–0.20**\* \| –0.10 | **0.17**\* \| 0.01 |
| Spital | 0.15 \| **0.23**\* | –0.12 \| –0.16 | **–0.30**\* \| **–0.24**\* | 0.14 \| 0.04 |
| Stift Bern | **0.23**\* \| **0.20**\* | **–0.15** \| **0.22** | **–0.19**\* \| –0.02 | **0.13** \| 0.04 |
| Töss | **0.15** \| 0.08 | **–0.19**\* \| –0.10 | –0.21 \| **–0.14**\* | 0.05 \| –0.07 |
| Trachselwald | **0.29**\* \| **0.20**\* | –0.07 \| –0.04 | –0.06 \| **–0.15**\* | 0.05 \| –0.03 |
| Wädenswil | **0.40**\* \| **0.11**\* | 0.01 \| **–0.20** | **–0.15** \| –0.09 | 0.00 \| –0.07 |
| Wangen | **0.25** \| **0.23**\* | –0.01 \| 0.07 | –0.09 \| –0.03 | 0.15 \| 0.05 |
| Zofingen | **0.20**\* \| **0.26**\* | –0.12 \| 0.09 | **–0.30**\* \| –0.08 | **0.13** \| 0.03 |
| *Canton Aargau mean tithe* | **0.32**\* \| **0.22**\* | **–0.18**\* \| **–0.22** | **–0.26** \| **–0.16**\* | 0.14 \| 0.10 |
| *Canton Bern mean tithe* | **0.21**\* \| **0.18**\* | **–0.18**\* \| **–0.15**\* | **–0.28** \| –0.23 | 0.12 \| 0.04 |
| *Canton Schwyz mean tithe* | **0.21** \| **0.33**\* | –0.09 \| 0.15 | **–0.20** \| –0.09 | 0.20 \| 0.04 |
| *Canton Vaud mean tithe* | **0.13** \| 0.08 | **–0.24**\* \| **0.20** | **–0.17**\* \| –0.07 | **0.14** \| –0.01 |
| *Canton Zurich mean tithe* | **0.32**\* \| **0.13**\* | **–0.14** \| **–0.21** | **–0.23** \| **–0.13**\* | 0.07 \| –0.07 |
| Yield ratio rye Basel | **–0.22**\* \| **–0.18**\* | **–0.19**\* \| –0.05 | **–0.23**\* \| **–0.23**\* | 0.03 \| **–0.13** |
| Yield ratio rye Winterthur | 0.03 \| 0.06 | **–0.18**\* \| –0.03 | **–0.18**\* \| –0.05 | **0.15** \| 0.05 |
| Yield ratio rye Zurich | –0.08 \| –0.07 | –0.06 \| 0.03 | 0.04 \| –0.05 | **0.16**\* \| 0.08 |
| Yield ratio spelt Basel | **–0.14**\* \| –0.07 | –0.10 \| –0.01 | **–0.19**\* \| **–0.19**\* | 0.06 \| –0.11 |
| Yield ratio spelt St. Gallen | –0.07 \| 0.01 | 0.00 \| 0.07 | 0.08 \| 0.02 | 0.04 \| –0.02 |
| Yield ratio spelt Winterthur | 0.02 \| 0.03 | **–0.17** \| –0.12 | **–0.17**\* \| –0.07 | **0.22**\* \| 0.08 |
| Yield ratio spelt Zurich | 0.06 \| 0.06 | –0.05 \| **0.14** | –0.09 \| –0.06 | 0.11 \| –0.02 |
| *Yield ratio rye average* | **–0.15**\* \| **–0.12**\* | **–0.22**\* \| –0.08 | **–0.15**\* \| –0.14 | **0.17**\* \| –0.04 |
| *Yield ratio spelt average* | –0.09 \| 0.01 | –0.10 \| 0.03 | **–0.18**\* \| **–0.14**\* | **0.15**\* \| –0.04 |





**Table A5.** Climate–harvest correlations for Spain. Results using high-pass filtered data shown to the left of ' | ' and results using linearly detrended data to the right. Bold values indicate correlations significant at the $p = 0.05$ level with a $t$-test and an asterisk (*) significant correlations with the phase-scrambling test. All correlations for Spain, and their significance, are shown in Fig. 5.

| Dataset | $T_{MAM}$ | $T_{annual}$ | $P_{MAM}$ | $P_{annual}$ |
|---|---|---|---|---|
| Tithe Mondonedo | **–0.15**\* \| 0.12 | **–0.25**\* \| 0.09 | –0.05 \| –0.08 | 0.00 \| –0.01 |
| Tithe Santiago | –0.02 \| –0.10 | 0.03 \| –0.08 | –0.10 \| **–0.24**\* | –0.09 \| 0.01 |
| Tithe Galice | –0.05 \| –0.04 | –0.03 \| –0.04 | –0.12 \| **–0.19**\* | –0.06 \| –0.09 |
| Tithe Orense | 0.08 \| **0.17**\* | 0.04 \| **0.18**\* | –0.02 \| –0.05 | –0.02 \| –0.15 |
| Tithe wheat Aragon | 0.06 \| 0.03 | 0.12 \| –0.01 | 0.04 \| –0.07 | –0.01 \| **0.18** |
| Tithe non-wheat Aragon | 0.15 \| **0.23**\* | **0.20** \| **0.20**\* | 0.02 \| –0.01 | 0.09 \| 0.07 |
| Tithe barley Guadalajara | 0.21 \| 0.15 | 0.10 \| 0.15 | –0.17 \| **–0.29**\* | –0.28 \| 0.08 |
| Tithe oats Guadalajara | **0.41**\* \| **0.35**\* | **0.31**\* \| **0.38**\* | **–0.31**\* \| **–0.30**\* | –0.21 \| –0.07 |
| Tithe rye Guadalajara | 0.20 \| 0.09 | 0.06 \| 0.06 | 0.01 \| **–0.30**\* | –0.08 \| –0.11 |
| Tithe wheat Guadalajara | 0.12 \| –0.05 | 0.08 \| –0.05 | –0.05 \| **–0.24** | **–0.24** \| **–0.22** |
| Tithe total Guadalajara | 0.18 \| 0.04 | 0.11 \| 0.04 | –0.10 \| **–0.30**\* | **–0.26** \| –0.24 |
| Tithe wheat Calañas | **–0.24**\* \| –0.11 | **–0.31**\* \| –0.11 | **0.15** \| **0.15** | **0.18** \| **0.21**\* |
| Tithe wheat Puebla de Guzman | –0.13 \| –0.13 | **–0.20**\* \| **–0.15**\* | **0.19** \| **0.18**\* | **0.25**\* \| **0.19**\* |
| Tithe wheat Lucena | –0.01 \| –0.12 | –0.01 \| –0.13 | 0.04 \| –0.14 | 0.07 \| **–0.14** |
| Tithe wheat Bollullos | **–0.21**\* \| **–0.21**\* | –0.13 \| **–0.17**\* | –0.01 \| –0.12 | –0.02 \| –0.09 |
| Yield ratio wheat Farms of Matallana | –0.09 \| –0.16 | **–0.20** \| **–0.25**\* | **0.19** \| –0.01 | 0.05 \| –0.06 |
| Yield ratio wheat Quintanajuar farm | 0.12 \| 0.10 | 0.03 \| 0.03 | –0.07 \| –0.13 | –0.11 \| 0.02 |
| Yield per unit area wheat Panguia farm | –0.17 \| **–0.20** | **–0.24**\* \| **–0.25**\* | 0.19 \| 0.09 | **0.23** \| 0.08 |
| Yield ratio wheat La Burguilla farm | **–0.29**\* \| **–0.26**\* | –0.22 \| **–0.27**\* | **0.32**\* \| 0.15 | 0.16 \| 0.12 |
| Yield ratio wheat El Rincón farm | **–0.34**\* \| **–0.18**\* | –0.18 \| –0.10 | **0.23**\* \| 0.15 | 0.11 \| 0.04 |
| Yield ratio wheat Madrigalejo farm | **–0.30**\* \| **–0.24**\* | **–0.37**\* \| **–0.25**\* | **0.26**\* \| –0.06 | 0.11 \| –0.11 |
| Yield ratio wheat La Vega farm | –0.15 \| –0.18 | –0.17 \| –0.14 | 0.14 \| 0.02 | –0.12 \| –0.15 |
| Yield per unit area wheat Rinconada Alta farm | **–0.28**\* \| **–0.19** | **–0.37**\* \| **–0.25**\* | 0.17 \| –0.01 | 0.20 \| 0.05 |



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
