# Peer review of "Climatic signatures in early modern European grain harvest yields"

_Climate of the Past, 2022_

## Author Response (AR1)

**The Editor,** *Climate of the Past*

Fredrik Charpentier Ljungqvist
Professor of History, especially Historical Geography

Department of History *and*
Bolin Centre for Climate Research
Stockholm University
SE-106 91 Stockholm
Sweden

E-mail: fredrik.c.l@historia.su.se
Mobile phone: +46706620728

Monday, July 10, 2023

**Manuscript revision**

Dear editor Prof. Denis-Didier Rousseau,

Thank you very much for the positive evaluation of our manuscript entitled "Climatic signatures in early modern European grain harvest yields". We thank the two reviewers for their extensive and profound reviews and we are very grateful for the comments and suggestions. We have applied changes following the suggestions by the two reviewers and added more information to improve the manuscript. In particular, we have shortened the Results and Discussion sections, as one of the reviewers asked for, and we have to this end also removed the entire "Other region" section. A point-by-point response to the two reviewers' comments follows below.

We hope that the revised version of the manuscript will make the article suitable for publication in *Climate of the Past*.

Thank you very much and we are looking forward to hearing from you soon.

On behalf of the authors, yours sincerely,

Fredrik Charpentier Ljungqvist

**Point-by-point response to the reviewers' comments**

**Reviewer #1**

**Comment:** The issue of representativeness in space is my first question. The paper has selected several sites to indicate the harvest in Sweden, Switzerland, and Spain. Could the authors give more explanations how these sites are representative to show the harvest conditions of these three countries?

**Reply:** We only have data from the areas shown in Fig. 1. Note that for Sweden and Switzerland we use county and canton level data which should be more representative than point data. The choice to focus on Sweden, Switzerland, and Spain is made for two reasons: (1) most yield (ratio) series and tithe series are available from these countries for the early modern period, and (2) these countries, representing a north-to-south gradient in Europe, are representative for the climate–harvest relationship in northern, central and southern Europe, respectively. For both Sweden, Switzerland, and Spain the harvest series come from major agricultural regions. Thus, we consider them representative for the harvest conditions in the three countries although we note a large environmental heterogeneity within all three countries. This is explicitly now pointed out in the revised manuscript.

**Comment:** Second, I understand the data processing before the statistical analysis, such as detrend and smooth. I just wonder how large is the difference between the results raw data and processed data? If the authors use raw datasets, it will be also interesting to show human factors behind the climate-harvest linkage.

**Reply:** As seen in the correlation results, there is a dependence on the type of detrending and filtering applied – hence, the results from 'raw data' are expected to also yield differing results. We do not want to look at raw data as the low-frequency variability in harvest data is most likely influenced by factors more or less unaffected by climate (e.g., labour force availability, market access changes, wars etc.). That said, we actually did perform exploratory data analysis, at an early stage of our work, using 'raw' data for some data subsets. This yielded, in cases without a strong long-term trend, results rather similar to when using linearly detrended data. Please also see Fig. 2.

**Comment:** Third, the harvest conditions are investigated by two indicators, tithe and yield ratio, which have improved the findings currently made only according to yield ratios. However, I would suggest the authors to add more explanations on these two indicators to which extent they could be compared with each other.

**Reply:** We think that we actually discuss this issue rather extensively both in the Data and Method Section and in the Discussion Section. Furthermore, we are not really sure what the reviewer refers to with "findings currently made only according to yield ratios". We have further, in section 4.4 in the revision, further discussed the differences between tithe and yield ratio at various spatio-temporal scales.

**Comment:** Fourth, the authors used Granger Causality Analysis. This is a method very useful to check the temporal patterns. Furthermore, the authors use correlation analysis as well. Is there any other method potentially suitable for the study on climate-harvest linkage?

**Reply:** A few other methods could have been used in addition to the correlation and Granger Causality analyses applied, such as multivariate regression, linear or nonlinear. Extraction of leading patterns of variability by use of Principal Component Analysis (PCA) could also be applied, and if there are episodic excursions in the climate these would be expected to impact harvest data and could thus be revealed by so-called Superposed Epoch Analysis (SEA). The article is already long as we have thus not added such additional analyses. SEA can well be applied in situations where, e.g., volcanic forcing events are expected

to impact climate and/or climate-dependent things such as harvests, and we are working on such an analysis for another (unrelated) article at the moment.

**Comment:** Fifth, I am very curious about the implication from your study on the past to current societies under the warming threat. May I suggest the authors to share their views on the practical implications to modern era?

**Reply:** Modern agricultural practices are different from what was used during our study period (c. 1500–1800). We choose these limits in order to have many overlapping records as well as avoiding the period after fertilisation, mechanisation and new seed types became widespread, as this insulates harvest yields (and hence tithes) from climate and weather excursions to some extent. The modern climate–harvest relationship can be expected to be different from the one prior to 1800. We have now added, in the revised article, a new paragraph ending section 4.5 addressing this. A few lines have also been added to the Introduction.

**Reviewer #2**

**Comment:** This study is an admirable and important work; it presents a tremendous effort of all authors to find out the various kinds of data, put all data together, clean data, homogenize different formats of data, and explore statistical methods for analysis, not to mention that the data contains a huge variety in the attributes, sources, languages, and configurations. The authors also give detailed and comprehensive descriptions of the data profiles and the way they processed the data. They also exploited the statistical methods to justify the analysis and findings. I have no doubts on the scientific clarity and ambition the authors possessed to answer the longstanding research question of "the relationships between climate and grain harvest" which governs the very fundamental dimension of human security in history. However, I do have some comments and suggestions for the authors to further sharpen the paper structure and their findings.

**Reply:** Thank you so much for your positive evaluation of our work and for your comments and suggestions that we will address below.

**Comment:** The former part of the paper (approximately from introduction to data and methods) is clear and concise. The results and especially discussion parts are quite heavy for reading and contain a lot of details that may make sense but are not necessary for the readers. The authors are very knowledgeable on the climate and agriculture in Europe and spend a lot of extra space to explain possible sources for moderate to weak relationships found between climate and grain harvest in the three case countries, i.e., Sweden, Switzerland, and Spain. The explanations may be supported by previous literature, but they are speculation in nature, too detailed and lengthy. Many of them also look like authors' subjective interpretation of the results. In other words, no matter how the results are, there is always a good explanation for it. To make the results and discussions more readable and focused, I suggest the authors constrain the content to simply present what the statistics reveals from the data, cutting off the lengthy part of over-explanations. And I also see the benefits for authors to slightly restructure the paper. The reason is provided in the next point.

**Reply:** We have now both shortened, reworked and streamline the Results and Discussion sections in the revised article. In particular, we have removed all material related to the "Other Regions" and thus only keep the results regarding Spain, Switzerland and Sweden. This has considerably shorten the article as well as made it easier to read and follow.

**Comment:** One of the main reasons to make the paper lengthy is because the authors seem to be unsatisfied with the low correlations between climate factors and grain harvest. Therefore, they exploited every possible source of uncertainties in the grain harvest data (e.g., systematic or recording error in grain data, small geographical region not representative for the country, heterogeneous of the grain types, microclimate influence) for the explanation. As a matter of fact, not only the grain harvest data but also the palaeoclimate data all have their huge sources of uncertainties and are all very different in the formats and methodologies. Hence, considering all these messiness and heterogeneity, it is already quite

remarkable that statistical analysis can find a convincing result to explain the correlations between certain climate factors and grain harvest. I would say what this study discovers are the first-order, maybe not the most critical but the most common, climate factors relating to grain harvest, which is summer soil moisture in Sweden, winter temperature and precipitation in Switzerland, and more heterogeneous in Spain.

**Reply:** We have focused more on the first-order results in the revised article. However, we think that some explanations in the discussion of the type that we do have are useful. The Discussion section is substantially shorter now.

**Comment:** Following the previous point, I suggest the authors restructure the paper, explaining all data uncertainties in the material and methods section, which would then allow them to focus on the statistical results of the analysis and the implications of them from the scientific perspective, not from the data perspective. Aside from the hard work on data integration, a very important contribution of the paper is that it tells us (with sound justification) climate can play a role in the grain production and harvest but the importance (explanatory power) of the role and its influence can vary in different locations and at different times, controlled by other random and/or societal factors. Since this study is the first one, in my limited knowledge, to integrate the most accessible grain data in Europe, I would foreseen more studies in the near future to conduct various analysis including spatiotemporal analysis to further sort out, for example, spatiotemporal changes of different grain types and their separate relationships with climates. Of course, more research ideas can be stimulated by this study.

**Reply:** Thank you for this constructive comment. We have, in part, restructured the revised article and, as mentioned above, shorten the article.

**Comment:** Figure 1 is not mentioned in the main text.

**Reply:** Fixed.

**Comment:** In Figure 2, the chart mark of C & D needs to be corrected according to the caption.

**Reply:** Fixed.

**Comment:** In Table 1, is the number in the 'Gaps' column correct? The gaps show the percentage of missing values but the number varies a lot from 0.59 to 50.90, for example.

**Reply:** Yes, the percentage of missing values varies a lot. Barley and rye were the main two crop types in Sweden during the period and for them we have rather high coverage with limited gaps. For wheat and oats, on the other hand, we often only have certain periods with information as they – even together – stood for far less than 10% of the crop production and were not always counted in the tithes as they were too marginal.

**Comment:** When the percentage of missing value in a dataset is relatively large, e.g. more than ¼ of data series in Swedish grain tithe data sets has more than 20% of gaps, then using kernel smoothing to restore/fill data and using 10-year high-pass Gaussian (line 202) is rather smoothened method and can be a huge bias and a big source of noise. Do you consider giving up the data sets having a high percentage of missing values so that data analysis can be more responsive to reality and be more sensitive to the climate data?

**Reply:** The gaps are only filled by the kernel smoothing to deal consistently with the problems near the edges of the gaps. The gaps are re-inserted after the high-pass filtering and are NOT used in the later analysis. When doing time-filtering special care is always needed near the edges. This has now been made clearer.

**Comment:** I agree that correlation is not causation (line 236; old version). Therefore, although 'causality test' can test whether A happens and then B happens, it still can't indicate A is the cause of B. Because

there can be another factor C dominating whether A happens and then B can happen. So, this test only tells us the possible sequential correlation between A and B.

**Reply:** Yes, we agree that Granger 'causality test' only really inform us about the sequential correlation (the order). We have highlighted this even further in the revised article. The point of the Granger test is to ensure that we find the situations where A 'causes' B **but not the opposite**, as the occurrence of B causing A would clearly indicate absence of any believable causation.

**Comment:** line 532–535 (old version), to explain why the grain price was significantly lower when temperature was higher in Europe, the authors propose some points quite confusing: inferior grain quality can cause lower price in wetter and cooler conditions (this does not explain the point). In the climate-crop price study in China, it is generally found that when the weather is appropriate and there is good harvest, the crop price tends to be lower due to excessive food supply.

**Reply:** We have heavily reworked and shortened this entire section in the revised version.

**Comment:** The most vital technical comments I like to address here is the authors' interpretation of the detrended data (in many places but mostly in line 560-580; old version). I agree that high-pass filtered data reflects high frequency information; however, linearly detrended data hardly can reflect low-frequency information from my knowledge. The 'detrended' data is the data by removing the fitted line – the trend – so the most fundamental purpose to use detrended data is to see the data variability associated with a particular time scale. In other words, detrended data which focuses on variability can absolutely contain high frequency information. Although you might be able to say that compared to high-pass filtered data, detrended can reflect lower frequency trend and variability. But still this can be controversial and debated among scientists. If the purpose is to see the signals from high- and low-frequency, why not the authors use high-pass and low-pass filtered data for analysis? Moreover, in line 578, it is said that 'there are more uncertainties and sources of biases in the low frequency than in the high frequency domain'. This declaration can be controversial and dangerous. It is quite common in data science that we found a lot of (more) noises in the high frequency data (than low frequency). I think this sentence (and others of the kind) is not important in the study. Authors can remove those controversial viewpoints and make readers more focused on the important findings.

**Reply:** Linear detrending only removes some of the low-frequency variability. Any 'wiggles on scales shorter than the length of the data series' are retained to a large degree. Thus, linear detrending does not remove 'all low frequency variability' in the way e.g. high-pass filtration does (up to the band-pass, of course). It is certainly NOT the case that there are less uncertainties ("noise") in the harvest (tithe and yield ratio) data or in the palaeoclimate data at the lower frequencies than at the higher frequencies. Both tree-ring data and historical documentary data have their greatest strength at inter-annual to decadal frequency bands with well-known and much researched challenges at lower frequencies. The inter-series correlation is typically very high at high-frequency scales whereas it is often rather low at low frequencies. This is also the case for the harvest (tithe and yield ratio) data. They show a coherent pattern at high-frequency scales but sometimes even opposite long-term trends. Such issues are related to both how these series were collocated (as tax records as the tax base could vary over time) and due to local agricultural changes related to particular socio-economic and demographic conditions. The number of farmers, and the size of their holdings, that the records are based on did vary over time – thus, making the low-frequency information more unreliable.

**Comment:** There remains some minor issues regarding too detailed or over explanations of the analysis. All of those can be dealt with under the points 1) and 2).

**Reply:** We believe that this has been addressed now in the revision of the article.

**Comment:** Overall, this is an important paper much worthwhile publishing. I hope my comments can be helpful for the authors to further improve their rationale of the paper and to increase sharpness and scientific clarity.

**Reply:** Thank you again for your constructive and positive evaluation of our article.